# Activation of a neural stem cell transcriptional program in parenchymal astrocytes

Jens P Magnusson‡, Margherita Zamboni†, Giuseppe Santopolo†, Jeff E Mold, Mauricio Barrientos-Somarribas, Carlos Talavera-Lopez§, Björn Andersson, Jonas Frisén*

Department of Cell and Molecular Biology, Karolinska Institute, Stockholm, Sweden

**Abstract** Adult neural stem cells, located in discrete brain regions, generate new neurons throughout life. These stem cells are specialized astrocytes, but astrocytes in other brain regions do not generate neurons under physiological conditions. After stroke, however, striatal astrocytes undergo neurogenesis in mice, triggered by decreased Notch signaling. We used single-cell RNA sequencing to characterize neurogenesis by Notch-depleted striatal astrocytes in vivo. Striatal astrocytes were located upstream of neural stem cells in the neuronal lineage. As astrocytes initiated neurogenesis, they became transcriptionally very similar to subventricular zone stem cells, progressing through a near-identical neurogenic program. Surprisingly, in the non-neurogenic cortex, Notch-depleted astrocytes also initiated neurogenesis. Yet, these cortical astrocytes, and many striatal ones, stalled before entering transit-amplifying divisions. Infusion of epidermal growth factor enabled stalled striatal astrocytes to resume neurogenesis. We conclude that parenchymal astrocytes are latent neural stem cells and that targeted interventions can guide them through their neuronal differentiation.

**\*For correspondence:**
jonas.frisen@ki.se

†These authors contributed equally to this work

**Present address:** ‡Department of Bioengineering, Stanford University, Stanford, United States; §The Francis Crick Institute, London, United Kingdom

**Competing interests:** The authors declare that no competing interests exist.

## Introduction

Neurogenesis is extremely limited in the adult brain. In most mammals, specialized astrocytes in the subventricular zone (SVZ) and hippocampal dentate gyrus (DG) are stem cells and generate new neurons continuously, but apart from that, the brain's ability to replace lost neurons is very limited. However, in response to an experimental stroke or excitotoxic lesion, some astrocytes in the mouse striatum can generate neurons (*Magnusson et al., 2014*; *Nato et al., 2015*). This neurogenic response is regulated by Notch signaling and can be activated even in the uninjured mouse striatum by deleting the Notch-mediating transcription factor *Rbpj* (*Magnusson et al., 2014*). Striatal astrocytes undergo neurogenesis by passing through a transit-amplifying cell stage. But it is not known whether these astrocytes become bona fide neural stem cells. If they do, this could have far-reaching implications for regenerative medicine. Astrocytes make up a large fraction of all brain cells (10–20% in mice) (*Sun et al., 2017*) and are distributed throughout the central nervous system. They would thus represent a very abundant source of potential neural stem cells that might be recruited for therapeutic purposes.

Although certain injuries and *Rbpj* deletion can both trigger neurogenesis by astrocytes, it almost exclusively does so in the striatum. And even within the striatum, primarily the astrocytes in the medial striatum readily activate neurogenic properties (*Figure 1a*). This suggests that neurogenic parenchymal astrocytes either occupy an environmental niche favorable to neurogenesis or that only they have an inherent neurogenic capacity. In order to recruit astrocytes for therapeutic neurogenesis, a first step is to understand the mechanisms underlying this process. If these mechanisms are

**eLife digest** Regenerative medicine aims to help the body replace damaged or worn-out tissues, often by kick-starting its own intrinsic repair mechanisms. However, the brain cannot easily repair itself, and therefore poses a much greater challenge. This is because nerve cells or neurons, which underpin learning, memory, and many other abilities, are also the brain's greatest weakness when it comes to tissue repair.

In most parts of the adult brain, neurons are never replaced after they die. This means that damage to brain tissue – for example, after a stroke – can have severe and long-lasting consequences. Neural stem cells are one type of brain cell that can turn into new neurons if needed, but they are only found in a few parts of the brain and cannot fix damage elsewhere.

More recent work in mice has shown that astrocytes, a common type of support cell in the brain that help keep neurons healthy, could also generate new neurons following a stroke. However, the ability was restricted to small numbers of astrocytes in a specific part of the brain. Here, Magnusson et al. set out to determine the molecular mechanisms behind this regenerative process and why it is unique to certain astrocytes.

The researchers used a technique called single-cell RNA sequencing to analyze the genetic activity within individual mouse astrocytes that had been exposed to conditions mimicking a stroke. This method revealed which genes are switched on or off, thus generating a profile of gene activity for each astrocyte analyzed.

This experiment showed that the profiles of astrocytes that had started to produce neurons were in fact nearly identical to neural stem cells. Even the astrocytes that could not generate neurons took the first few steps towards this genetic state; however, they stalled early in the process. Treating the brains of mice withepidermal growth factor, a powerful molecular signal that stimulates cell growth, kick-started nerve cell production in a subset of these cells – showing that at least some of the non-regenerative astrocytes could be stimulated to make neurons if given the right treatment.

The results of this study shed new light on how some astrocytes in the brain gain the ability to form new neurons. In the future, this knowledge could help identify a source of replacement cells to regenerate the injured brain.

understood, they could potentially be targeted to induce localized therapeutic neurogenesis throughout the central nervous system.

Here, we generated two separate single-cell RNA sequencing datasets to study neurogenesis by parenchymal astrocytes in mice. We found that, at the transcriptional level, *Rbpj*-deficient striatal astrocytes became highly similar to SVZ neural stem cells and that their neurogenic program was nearly identical to that of stem cells in the SVZ, but not the DG. Interestingly, astrocytes in the non-neurogenic somatosensory cortex also initiated a neurogenic program in response to *Rbpj* deletion, but all stalled prior to entering transit-amplifying divisions and failed to generate neuroblasts. In the striatum, too, many astrocytes halted their development prior to entering transit-amplifying divisions. We found that stalled striatal astrocytes could be pushed into transit-amplifying divisions and neurogenesis by an injection of epidermal growth factor (EGF), indicating that it is possible to overcome roadblocks in the astrocyte neurogenic program through targeted manipulations. Taken together, we conclude that parenchymal astrocytes are latent neural stem cells. We posit that their intrinsic neurogenic potential is limited by a non-permissive environment. Recruiting these very abundant latent stem cells for localized therapeutic neurogenesis may be possible but is likely to require precise interventions that guide them through their neurogenic program.

## Results

### Transcriptome-based reconstruction of neurogenesis by striatal astrocytes

To understand the cellular mechanisms underlying neurogenesis by parenchymal astrocytes, we decided to perform single-cell RNA sequencing of striatal astrocytes undergoing neurogenesis in vivo. To this end, we generated two separate single-cell RNA sequencing datasets, each with

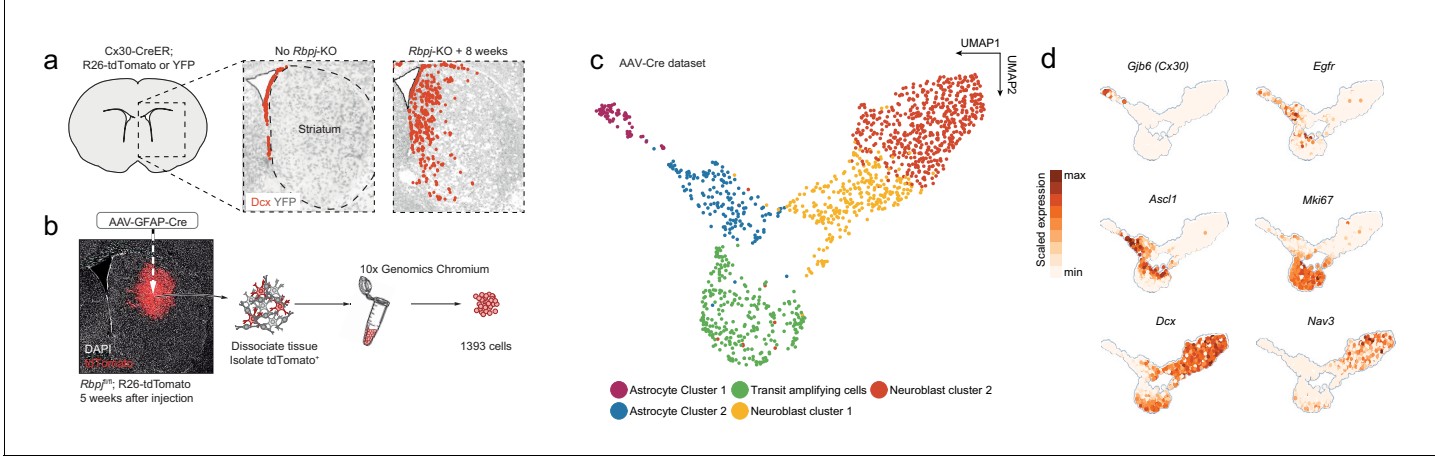

**Figure 1.** Neurogenesis by striatal astrocytes can be reconstructed using single-cell RNA sequencing. (a) Deletion of the gene encoding the Notch-mediating transcription factor *Rbpj* activates a latent neurogenic program in striatal astrocytes (*Magnusson et al., 2014*). Nuclei of Dcx⁺ neuroblasts are indicated by red dots. Not all striatal astrocytes undergo neurogenesis, shown by the restricted distribution of Dcx⁺ neuroblasts and the fact that many recombined astrocytes (gray) remain even 2 months after *Rbpj* deletion. (b) We performed single-cell RNA sequencing using two protocols. For the AAV-Cre dataset, we deleted *Rbpj* exclusively in striatal astrocytes using a Cre-expressing AAV and sequenced the transcriptomes of recombined cells five weeks later. (c) Dimensionality reduction using UMAP captures the progression from astrocytes, through proliferating transit-amplifying cells, to neuroblasts. Panel (d) shows markers for the different maturation stages.

The online version of this article includes the following figure supplement(s) for figure 1:

**Figure supplement 1.** Cell census from AAV-Cre dataset after local recombination of striatal cells.

complementary strengths. For the first dataset, which we call the Cx30-CreER dataset, we used Connexin-30 (Cx30; official gene symbol *Gjb6*)-CreER mice to delete *Rbpj* and activate heritable *tdTomato* or *YFP* expression in astrocytes throughout the brain. This method targeted up to 100% of astrocytes in the striatum (tdTomato expression in 92 ± 10% of glutamine synthetase⁺, S100⁺ cells; mean ± S.D.; *n* = 6 mice). These mice also target neural stem cells in the SVZ, but no other brain cell types (*Magnusson et al., 2014*). *Rbpj* deletion in healthy mice was chosen as the means by which to activate the neurogenic program of astrocytes. That is because (1) it constitutes a single, precisely timed trigger throughout the brain, (2) it mimics the endogenous stimulus of reduced Notch signaling by which stroke induces neurogenesis by astrocytes (*Magnusson et al., 2014*), and (3) it does not induce potentially confounding cellular processes like reactive gliosis or cell death. Striatal tdTomato⁺ cells were isolated by flow cytometry from mice homozygous for the *Rbpj* null allele 2, 4 and 8 weeks after tamoxifen administration, and from control mice with intact *Rbpj*, 3 days after tamoxifen administration (hereafter called ground-state astrocytes) (*Figure 2—figure supplement 1a–d*). Sequencing libraries were prepared using Smart-seq2 (*Picelli et al., 2013*).

In addition to parenchymal astrocytes, Cx30-mediated recombination also targets SVZ neural stem cells, whose progeny might be sorted along with striatal astrocytes and, thus, confound our results. We therefore generated a second single-cell RNA sequencing dataset, for which labeling of cells was restricted to the striatum. Striatal astrocytes were selectively recombined via injection of an adeno-associated virus (AAV) expressing Cre under the control of the astrocyte-specific GFAP promoter into the striatum of *Rbpj*ᶠˡ/ᶠˡ; R26-loxP-Stop-loxP-tdTomato mice (*Figure 1b*). We isolated tdTomato⁺ cells 5 weeks after virus injection, a time point at which Ascl1⁺ astrocytes, transit-amplifying cells and neuroblasts are all present (*Magnusson et al., 2014*). We sequenced these cells using 10X Chromium by 10X Genomics. Despite the GFAP promoter, the AAV-Cre dataset also contained oligodendrocytes and some microglia (*Figure 1—figure supplement 1*). These were excluded from the final dataset based on our previous finding that they are not involved in a neurogenic program (*Magnusson et al., 2014*). The AAV-Cre dataset contained a somewhat higher proportion of transit-amplifying cells than the Cx30-CreER dataset (*Figure 2—figure supplement 1f*), likely because the AAV-Cre dataset only contained cells from 5 weeks after *Rbpj*-KO, near the peak of transit-amplifying divisions (*Magnusson et al., 2014*).

We first asked whether we could reconstruct striatal astrocyte neurogenesis computationally using the AAV-Cre dataset. Using Uniform Manifold Approximation and Projection (UMAP), we found that the sequenced cells indeed included *Rbpj*-deficient astrocytes (*Cx30*⁺), activated astrocytes and transit-amplifying cells (*Egfr*⁺, *Ascl1*⁺, *Mki67*⁺, *Dcx*⁺) and neuroblasts (*Dcx*⁺), up until the migratory neuroblast stage (*Nav3*⁺) (*Figure 1c–d*). We conclude that it is possible to use single-cell RNA sequencing to study the transcriptional mechanisms underlying the astrocyte neurogenic program.

## Neurogenesis by astrocytes is dominated by early transcriptional changes associated with metabolism and gene expression

We were interested in the transcriptional changes that take place as astrocytes first initiate the neurogenic program. For this question, we primarily used the Cx30-CreER dataset, which included astrocytes both at ground state and at 2, 4 and 8 weeks after *Rbpj* deletion (*Figure 2a*). Furthermore, the Cx30-CreER mice allowed us to study the neurogenic process in the uninjured brain, without the potentially confounding effects of a needle injection. Two weeks after *Rbpj* deletion, astrocytes had upregulated genes mostly related to transcription, translation and metabolism (*Figure 2b*, S1g, *Supplementary file 1*). To study in detail how gene expression changed over the course of neurogenesis, we next reconstructed the differentiation trajectory of these astrocytes computationally using Monocle (*Trapnell and Cacchiarelli, 2014*; *Figure 2—figure supplement 2*; Methods). This analysis captured the progression from astrocyte to neuroblast in pseudotime (*Figure 2c*). The pseudotemporal axis reflected neurogenesis, as revealed by plotting along pseudotime the expression of genes that are dynamically induced in canonical neurogenesis (*Figure 2d*). This analysis confirmed the previous finding that neurogenesis by parenchymal astrocytes involves transit-amplifying divisions (*Magnusson et al., 2014*; *Niu et al., 2013*), and does not occur through direct transdifferentiation, a process during which intermediate stages of differentiation are skipped (*Briggs et al., 2017*).

In order to better understand the transcriptional changes that take place as striatal astrocytes initiate neurogenesis, we used Monocle's gene clustering algorithm, which clusters genes whose expression levels vary similarly along pseudotime (*Figure 2e*). These gene clusters were then characterized using the gene ontology (GO) tool DAVID (*Huang et al., 2009*). This approach enabled us to see which biological processes changed as differentiation proceeded (*Supplementary file 2*). We found that, as astrocytes initiated their neurogenic program, genes associated with metabolism, particularly lipid (e.g. *Fabp5*, *Fasn*, *Elovl5*) and carbohydrate metabolism (e.g. *Gapdh*, *Aldoa*, *Aldoc*) were at first upregulated but dramatically downregulated as cells entered transit-amplifying divisions (*Figure 2e–f*, [*Cluster 1*]). This suggested that the neurogenic program requires high metabolic activity to prepare for transit-amplifying divisions. In addition to these metabolic genes, the expression of genes involved in the regulation of transcription (e.g. mRNA splicing genes *Snrpb*, *Srsf5*, *Srsf3*) and translation (e.g. ribosomal subunits *Rpl17*, *Rpl23*, *Rps11*) progressively increased throughout the neurogenic process (*Figure 2e–f*, [*Cluster 2*]). Accordingly, we found that the number of genes detected per cell increased in the initial stages of neurogenesis (*Figure 2g*): pre-division astrocytes expressed 1.4 times as many genes as ground-state astrocytes (p=$10^{-12}$; 95% C.I. 1.3–1.5; Materials and methods). In theory, such an increase in the number of detected genes could be an artifact caused by a simultaneous increase in cell size; however, no cell size increase was observed during this early phase of neurogenesis, as measured by flow cytometer forward scatter (*Figure 2h*).

The changes to metabolism and gene expression activity described above were the two dominating transcriptional changes that occurred early in the neurogenic process of astrocytes. At the peak of metabolic gene expression, astrocytes initiated transit-amplifying divisions (*Figure 2e*; Materials and methods). Accordingly, they upregulated genes associated with cell division (e.g. *Ccnd2*, *Ccng2*, *Cdk4*; *Figure 2e–f*, [*Cluster 3*]). Finally, as they exited transit-amplifying divisions, cells upregulated classical marker genes of neuroblasts (e.g. *Cd24a*, *Dcx*, *Dlx1/2*, *Tubb3* [*βIII tubulin*]; *Figure 2e–f*, [*Cluster 3*]), indicative of early neuronal commitment and differentiation.

Interestingly, one group of genes showed a remarkable pattern of initial downregulation followed by upregulation as cells developed into neuroblasts (*Figure 2e–f* [*Cluster 4*], S4a). This gene group consisted mainly of genes encoding proteins localized to synapses (e.g. *Cplx2*, *Ctbp2*, *Shank1*, *Chrm4, Bdnf*). This suggested that synapse maintenance was compromised very early as astrocytes initiated neurogenesis, but also that some of the same genes are important for synapse maintenance in both astrocytes and neurons.

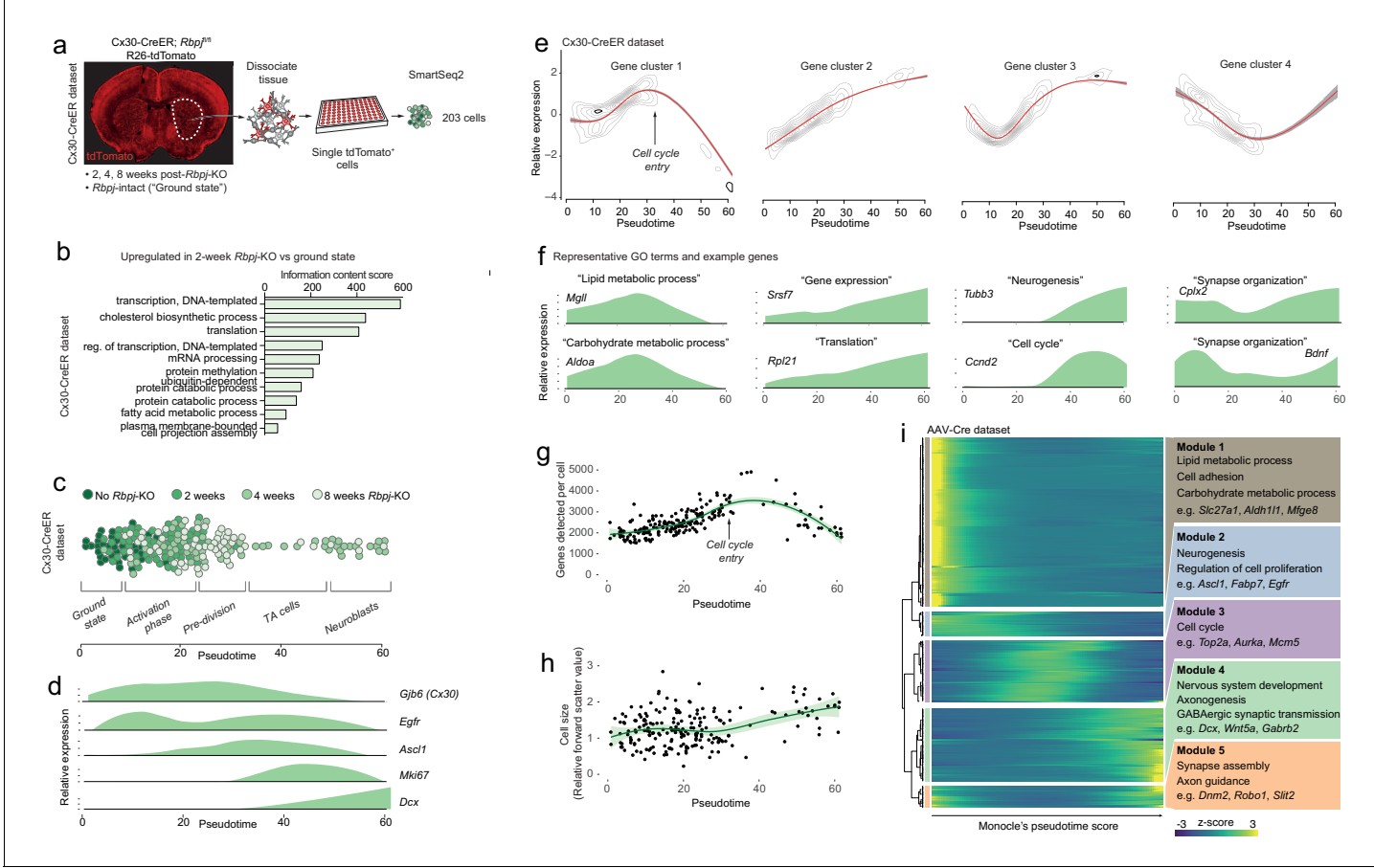

**Figure 2.** Neurogenesis by astrocytes is dominated by early transcriptional changes associated with metabolism and gene expression. (a) We prepared a second single-cell RNA sequencing dataset using SmartSeq2. This dataset consists of cells isolated from Cx30-CreER; *Rbpj*^fl/fl^ mice at 2, 4 and 8 weeks after *Rbpj* deletion, and ground-state astrocytes with intact *Rbpj*. (b) Two weeks after *Rbpj* deletion, astrocytes have upregulated genes mainly associated with metabolism and gene expression. (c) Dimensionality reduction using Monocle organizes astrocytes and their progeny along pseudotime (see *Figure 2—figure supplement 2* for cell state definitions). (d) Pseudotime captures the neurogenic trajectory of astrocytes, as shown by plotting the expression of classical neurogenesis genes along this axis. (e) Gene clustering and GO analysis shows distinctive dynamics of functional gene groups with example genes shown in (f). Genes involved in lipid and carbohydrate metabolism are upregulated initially but downregulated as cells enter transit-amplifying divisions. Genes involved in transcription and translation are expressed at steadily increasing levels. Accordingly, the number of genes detected per cell increases initially (g) – an increase not explained by a mere increase in cell size (h). A gene cluster with remarkable dynamics contains genes involved in synapse organization (e), presumably important for both astrocytes and neurons. (i) All these findings are supported by a pseudotemporal analysis of the AAV-Cre dataset. The AAV-Cre dataset, however, only contains cells from the 5-week time point and thus does not show changes that happen early in the neurogenic program. (TA cells, transit-amplifying cells).

The online version of this article includes the following figure supplement(s) for figure 2:

**Figure supplement 1.** Isolation of tdTomato^+^ cells for single-cell RNA sequencing and initial analyses.

**Figure supplement 2.** Hierarchical clustering of striatal astrocytes undergoing neurogenesis.

*Figure 2 continued on next page*

The gene expression changes described above were confirmed in the AAV-Cre dataset, using a differential expression analysis along pseudotime (*Figure 2i*). But because the AAV-Cre dataset only contained cells isolated 5 weeks after *Rbpj* deletion, this dataset did not contain ground-state astrocytes and could not capture changes that happened early in the neurogenic program.

## Astrocytes gradually lose astrocyte features as they enter the neurogenic program

Throughout the initial phase of neurogenesis and up until the point where cells entered transit-amplifying divisions, astrocytes maintained normal astrocyte morphology and expression of common

marker genes (e.g. *Aqp4*, *S100b*, *Slc1a2 [Glt1]*, *Glul*) (*Figure 2—figure supplement 3b–c*, *Video 1*). They never showed signs of reactive gliosis (e.g. morphological changes or upregulation of *Gfap*), indicating that reactive gliosis is not required for neurogenesis by astrocytes in vivo. As the astrocytes entered the neurogenic program, they did eventually lose their typical branched morphology; however, they did so only gradually, and only after entering transit-amplifying divisions. In fact, they retained rudiments of astrocytic processes for several transit-amplifying cell divisions (*Figure 2—figure supplement 3c*, *Video 1*). As an additional sign that they originated from parenchymal astrocytes, many clustered transit-amplifying cells retained trace levels of the astrocyte marker protein S100β, even though they no longer expressed the *S100b* gene. S100β protein is found in parenchymal astrocytes but not in SVZ stem cells (*Figure 2—figure supplement 3d–g*). Accordingly, we did not detect any S100β protein in SVZ transit-amplifying cells (*Figure 2—figure supplement 3d–g*). This suggests that the low levels of S100β protein seen in striatal transit-amplifying cells were lingering remnants of their parenchymal astrocyte origin. It indicates that trace levels of S100β protein can function as a short-term lineage-tracing marker for transit-amplifying cells derived from parenchymal astrocytes.

A recent study in *Drosophila melanogaster* showed that many quiescent neural stem cells are halted in the $G_2$ phase of the cell cycle (*Otsuki and Brand, 2018*). However, we analyzed EdU incorporation in dividing striatal astrocytes. We found some single astrocytes that had not yet divided but were EdU$^+$ (*Figure 2—figure supplement 3c*). This suggested that these astrocytes initiated the cell cycle by synthesizing DNA and thus had not been suspended in the $G_2$ phase.

## Parenchymal astrocytes are located upstream of a neural stem cell state and acquire a transcriptional profile of neural stem cells upon their activation

We were interested in a direct comparison between neurogenic parenchymal astrocytes and adult neural stem cells to see how their neurogenic transcriptional programs relate to one another. A number of published single-cell RNA sequencing datasets exist for mouse neural stem cells generated by 10X Chromium or Smart-seq2. We first analyzed our AAV-Cre dataset together with a previously published 10X Chromium dataset for SVZ neural stem cells and their progeny (*Zywitza et al., 2018*). A UMAP analysis of these cells together with our AAV-Cre dataset revealed that the striatal astrocytes and their progeny clustered closely together with the corresponding stages of SVZ cells (*Figure 3a–b*). This indicated that neurogenesis in these two brain regions proceeds through the same general developmental stages. Particularly interesting was the observation that our 'Astrocyte Cluster 2' grouped closely together with SVZ neural stem cells (*Figure 3b*), indicating that these cells are transcriptionally similar.

Because our AAV-Cre dataset only contained cells isolated 5 weeks after *Rbpj* deletion, this dataset could not reveal how striatal astrocytes in their ground state relate to SVZ stem cells. We therefore next turned to our Cx30-CreER dataset and analyzed these cells together with SVZ cells from another previously published Smart-seq2 dataset (*Llorens-Bobadilla et al., 2015*; *Figure 3c*). We used Monocle to perform pseudotemporal ordering of our Cx30-CreER dataset together with the SVZ cells, using genes involved in the GO term Neurogenesis (GO:0022008), and projected the cells onto the pseudotime axis (*Figure 3d*). Interestingly, on this neurogenic trajectory, ground state astrocytes were located upstream

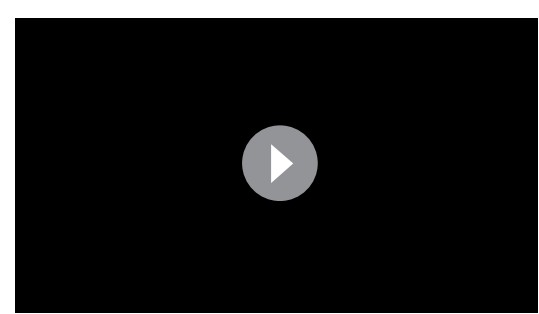

**Video 1.** Gradual loss of astrocyte morphology in astrocyte-derived transit-amplifying cells. Striatal astrocytes retain their branched astrocyte morphology (see YFP signal) even after they have initiated transit-amplifying divisions in vivo. Shown are several examples of different Ascl1$^+$ astrocytes and astrocyte-derived transit-amplifying cells at various stages of transit-amplifying divisions. EdU indicates that these cells had divided within the 2 weeks leading up to the analysis (Materials and methods). Note that astrocyte processes gradually become less branched and more retracted the longer transit-amplifying divisions proceed.
https://elifesciences.org/articles/59733#video1

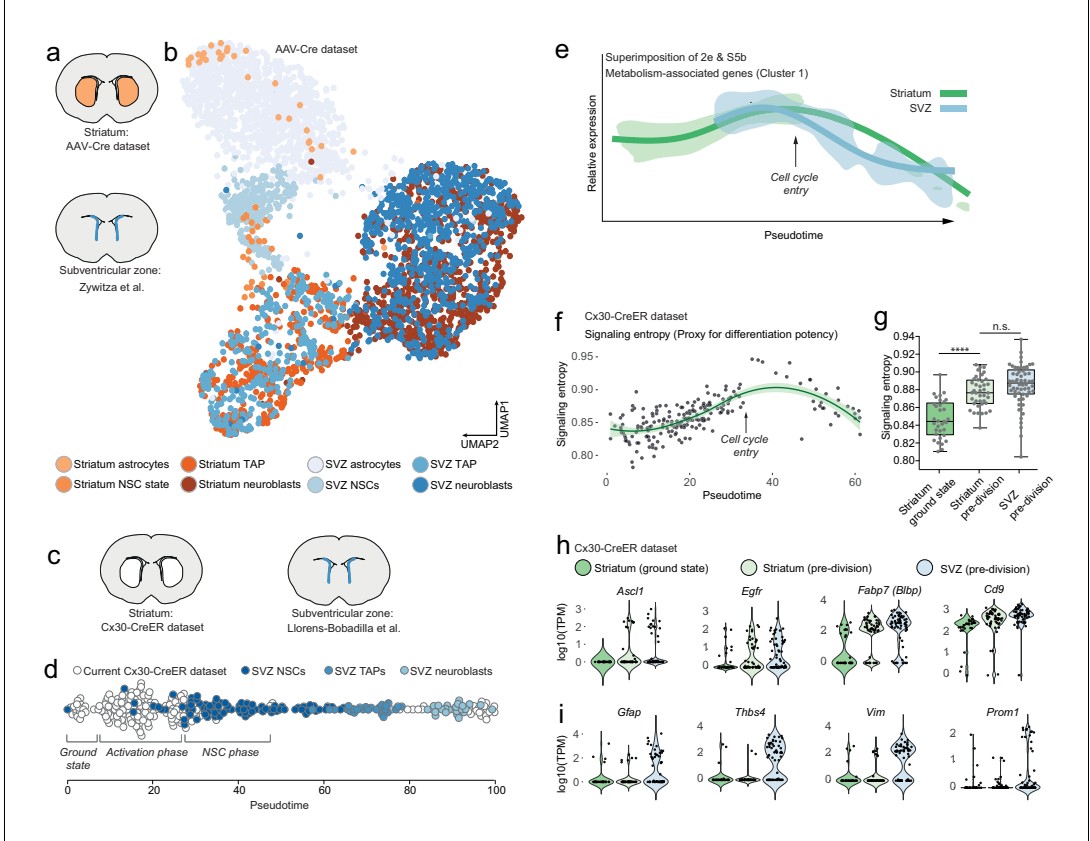

**Figure 3.** Striatal astrocytes are upstream of SVZ neural stem cells in the neurogenic lineage. (**a**) We analyzed our AAV-Cre dataset together with a previously published dataset of SVZ cells. (**b**) All maturational stages along the striatal neurogenic trajectory cluster together with the corresponding stages in the SVZ. The AAV-Cre dataset has few ground-state astrocytes because all cells were isolated five weeks after *Rbpj* deletion. (**c**) We therefore compared our Cx30-CreER dataset with a previously published dataset of SVZ cells using Monocle's pseudotime reconstruction (**d**). This analysis suggests that striatal astrocytes are located upstream of SVZ stem cells in the neurogenic lineage, as if representing highly quiescent neural stem cells that, upon *Rbpj* deletion, initiate a neurogenic program very similar to that of the SVZ stem cells. (**e**) Overlaying the expression pattern of metabolism-associated genes (compare *Figure 2e*, S5b) shows that striatal astrocytes initiate neurogenesis with a phase of metabolic gene upregulation, whereas SVZ stem cells permanently reside on this metabolic peak. (**f**) Signaling entropy, a reliable proxy for differentiation potency, increases in *Rbpj*-deficient astrocytes to reach the same levels as in SVZ stem cells (**g**). (**h**) Prior to entering transit-amplifying divisions, striatal astrocytes are in their most stem cell-like state, having upregulated many markers of neural stem cells (each dot is one cell. Note the bursting expression pattern of many of these genes). (**i**) Some classical markers, however, are not upregulated.

The online version of this article includes the following figure supplement(s) for figure 3:

**Figure supplement 1.** Gene clustering analysis of SVZ neural stem cells and their progeny from the dataset published by Llorens-Bobadilla et al.

of the SVZ stem cells, as if representing highly quiescent cells that pass through an initial activation phase before joining the SVZ stem cells (*Figure 3d*).

We next compared the gene expression changes that occur in SVZ cells with those in our Cx30-CreER dataset. We used Monocle's gene clustering analysis of only the SVZ cells, without the striatal cells. We first confirmed the previously published finding that SVZ neurogenesis is accompanied by large-scale changes in genes associated with metabolism (*Figure 3—figure supplement 1a–c*; *Llorens-Bobadilla et al., 2015*). Intriguingly, however, whereas SVZ neural stem cells appear to be maintained at a high baseline level of metabolic gene expression (*Figure 3—figure supplement 1b–c*), striatal astrocytes first passed through a phase of metabolic gene upregulation to reach this level, seen by superimposing the striatum-only (*Figure 2e*) and SVZ-only (*Figure 3—figure supplement 1b*) metabolic gene curves (*Figure 3e*). This suggested that the neurogenic program in striatal astrocytes starts with an activation phase during which these astrocytes begin to resemble SVZ stem cells with regard to metabolism-associated genes.

Similarly, evidence for an initial activation phase was found also in gene expression activity: As described above, we detected a 1.4-fold difference in the number of genes detected in *Rbpj*-deficient astrocytes compared to ground-state astrocytes. This difference is similar in magnitude to a previously detected 1.7-times difference between adult SVZ stem cells and striatal astrocytes (*Gokce et al., 2016*). This suggested that the gene expression activity of *Rbpj*-deficient astrocytes began to quantitatively resemble that of neural stem cells.

We asked if we could use our RNA sequencing data to address whether the differentiation potency of striatal astrocytes begins to resemble that of SVZ stem cells. 'Signaling entropy' is a measure of the interconnectedness of a cell's proteome. It represents the average number of possible protein-protein interactions by all proteins expressed by a cell. It is a reliable proxy for a cell's differentiation potency (*Teschendorff and Enver, 2017*; *Chen and Teschendorff, 2019*), the underlying rationale being that a highly interacting proteome enables promiscuous signaling, which in turn facilitates responding to a variety of differentiation signals. As a validation of the signaling entropy concept, we plotted signaling entropy versus pseudotime in the SVZ dataset and saw that SVZ cells exhibit their maximum signaling entropy as activated stem cells or transit-amplifying cells (*Figure 3—figure supplement 1d*). We found that the signaling entropy of striatal astrocytes increased after *Rbpj* deletion (*Figure 3f*), to the levels seen in SVZ stem cells (*Figure 3g*) (One-way ANOVA: $F = 35.62$, $p<0.0001$; Ground-state [$0.8467 \pm 0.0214$] vs. pre-division astrocytes [$0.8771 \pm 0.0171$, mean $\pm$S.D], $p<0.0001$ [unpaired t-test]; pre-division astrocytes vs. pre-division SVZ [$0.8848 \pm 0.0232$], $p=0.0721$). This supported the conclusion that striatal astrocytes achieve the same level of differentiation potency as SVZ neural stem cells.

Right before entering transit-amplifying divisions, astrocytes were in their most neural stem cell-like state. In addition to the similarities described above, striatal astrocytes in this pre-division state had upregulated some marker genes of neural stem cells (e.g. *Fabp7 [Blbp]*, *Egfr*, *Ascl1*, *Cd9*, *Cd81*, *Slc1a3 [Glast]*; *Figure 3h* and Figure 6). Other such markers were already expressed by ground-state astrocytes and their RNA levels did not change after *Rbpj* deletion (e.g. *Tnfrsf19 (Troy)*, *Sox2*, *Nr2e1 (Tlx)*, *Msi1 [Musashi-1]*, *Rarres2*; *Figure 3—figure supplement 1e*). Interestingly, some classical neural stem cell markers were never upregulated by *Rbpj*-deficient astrocytes. For example, *Gfap*, *Prom1*, *Thbs4* and *Vim* remained at low levels throughout the neurogenic process (*Figure 3i*), showing that neurogenic astrocytes do not become transcriptionally identical to SVZ stem cells. *Supplementary file 1* includes a full list of genes differentially expressed between ground-state astrocytes and those in their stem cell-like pre-division state.

## The neurogenic program of striatal astrocytes is more similar to that of SVZ stem cells than to that of DG stem cells

In the adult brain, active neural stem cells exist in two regions, the SVZ and the DG, and produce distinct neuronal subtypes. We asked whether the neurogenic program of striatal astrocytes is more similar to that of stem cells in the SVZ or the DG. For this, we integrated our AAV-Cre dataset with the aforementioned SVZ dataset from *Zywitza et al., 2018*. and another previously published 10X Chromium dataset of DG neurogenesis (*Hochgerner et al., 2018*; *Figure 4a*). We used our AAV-Cre dataset rather than the Cx30-CreER dataset for this comparison because previously published 10X Chromium datasets exist for the SVZ and DG that contain both stem cells, transit-amplifying cells and neuroblasts. A clustering analysis revealed that cells clustered together with other cells of the same maturational stage, regardless of which brain region they came from (*Figure 4a*). This indicates that neurogenesis proceeds along the same general steps in all three regions. To perform a more focused comparison of the neurogenic transcriptional programs in the striatum, SVZ and DG, we next performed pairwise correlation between striatal cells and the corresponding maturational stages in the SVZ and DG (*Figure 4b–c*). We found that the transcriptomes of the striatal cells were more similar to those of SVZ stem cells than to those of DG stem cells in every maturational stage, in particular at the transit-amplifying cell and neuroblast stages (*Figure 4b–c*).

We next took a more detailed look at neuroblasts, and only included transcription factors in our analysis (*Hu et al., 2019*), reasoning that the neuroblast transcription factor profile is indicative of commitment to neuronal subtypes. We performed three pairwise differential expression analyses between the neuroblasts in the striatum, SVZ and DG and plotted the most differentially expressed transcription factors in a heatmap (*Figure 4d*). This revealed the close transcriptional similarity between striatal and SVZ neuroblasts, suggesting that striatal astrocytes and SVZ stem cells are

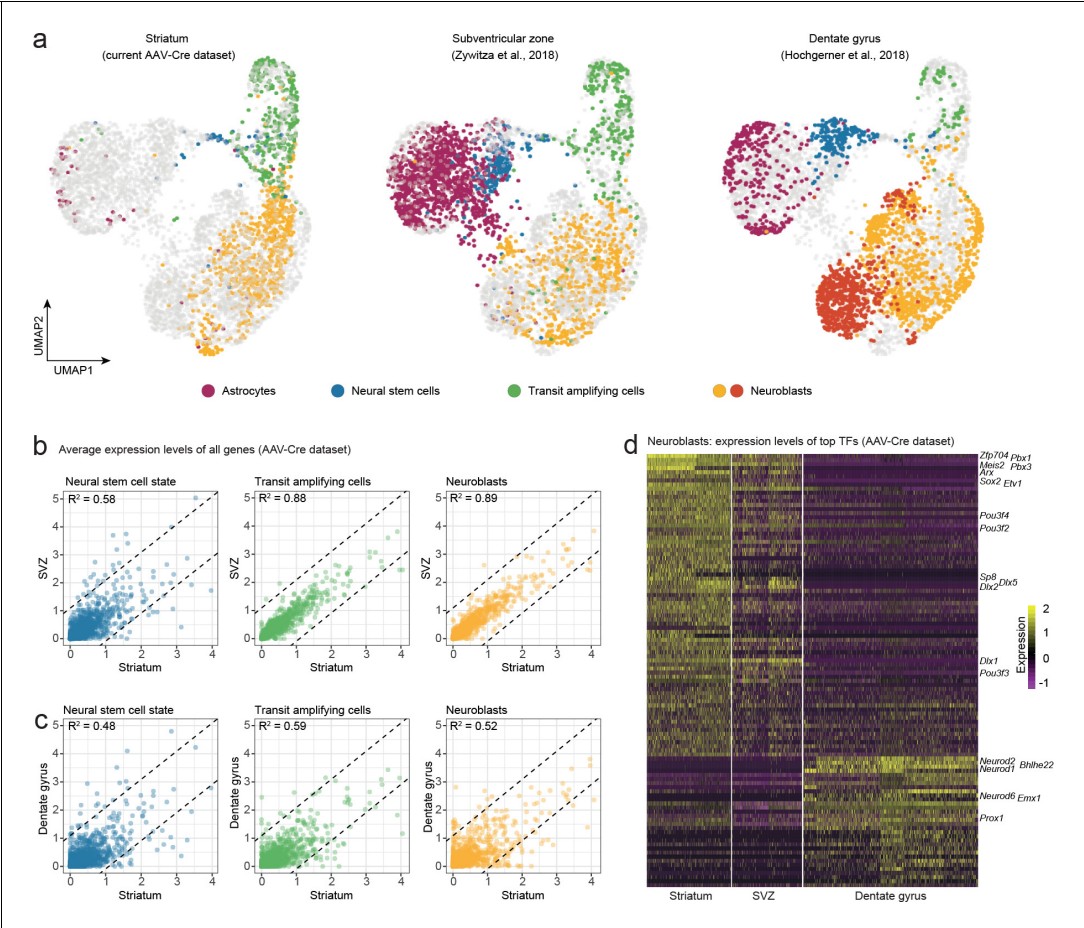

**Figure 4.** Neuroblasts generated by striatal astrocytes are more similar to SVZ neuroblasts than to DG neuroblasts. We analyzed our AAV-Cre dataset together with two previously published datasets from the SVZ and DG. (a) Cells from each maturational stage cluster together regardless of origin, highlighting that neurogenic cells transition through the same general stages in all three regions. (b) We performed pairwise comparisons between brain regions by plotting the average expression levels of all genes. Striatal astrocytes in the stem cell-like pre-division state are only moderately similar to SVZ neural stem cells (b; $R^2 = 0.58$) but become more similar as they develop into transit-amplifying cells (b; $R^2 = 0.88$) and neuroblasts (b; $R^2 = 0.89$; b), indicating that striatal astrocytes and SVZ stem cells converge on nearly identical neurogenic programs. (c) In contrast, stem cells in the DG and their progeny never acquire more than moderate similarity to the striatal cells. (d) A heat map shows some of the top transcription factors that define the similarities and differences among neuroblasts. This suggests that striatal astrocytes are primed to generate very similar neuronal subtypes as the SVZ stem cells.

predisposed to generate very similar neuronal subtypes (GABAergic interneurons), different from those in the DG (excitatory granule neurons). In summary, the transcriptional program governing neurogenesis by striatal astrocytes is highly similar, but not identical, to that of SVZ cells, and less similar to that of DG stem cells. We speculate that this similarity likely reflects a shared developmental history. It has, for example, been shown that the region-specific transcriptional signature among astrocytes is conserved when astrocytes are reprogrammed directly into neurons (*Herrero-navarro et al., 2020*).

## Astrocytes outside the striatum initiate an unsuccessful neurogenic program in response to *Rbpj* deletion

We previously found that, in addition to the striatum, *Rbpj* deletion triggers astrocyte neurogenesis in layer 1 of the medial cortex (*Magnusson et al., 2014*). Now, we asked whether astrocytes in other parts of the brain also show signs of activating a neurogenic program, even though they do not complete neurogenesis. To detect faint signs of an initiated neurogenic program, we first performed an immunohistochemical staining against the early neurogenesis-associated transcription factor Ascl1 in

Cx30-CreER; *Rbpj*^fl/fl^ mice, 4 weeks after tamoxifen administration. Importantly, we used tyramide-based signal amplification to boost the fluorescent signal in order to be able to detect low levels of Ascl1 protein. Intriguingly, this showed that many astrocytes throughout the brain had upregulated low levels of Ascl1 in response to *Rbpj* deletion (*Figure 5a–c*). This suggested that astrocytes in all brain regions initiate at least one early step in a neurogenic program in response to *Rbpj* deletion. To learn more, we next performed single-cell RNA sequencing on astrocytes isolated from the non-neurogenic somatosensory cortex of the same Cx30-CreER mice that we had used for the striatum analysis (*Figure 5d*). We chose the Cx30-CreER model for this analysis because it induces *Rbpj* deletion simultaneously throughout the brain, does not lead to any potentially confounding tissue damage, and does not suffer from viral tropism-induced differences in recombination efficiency between

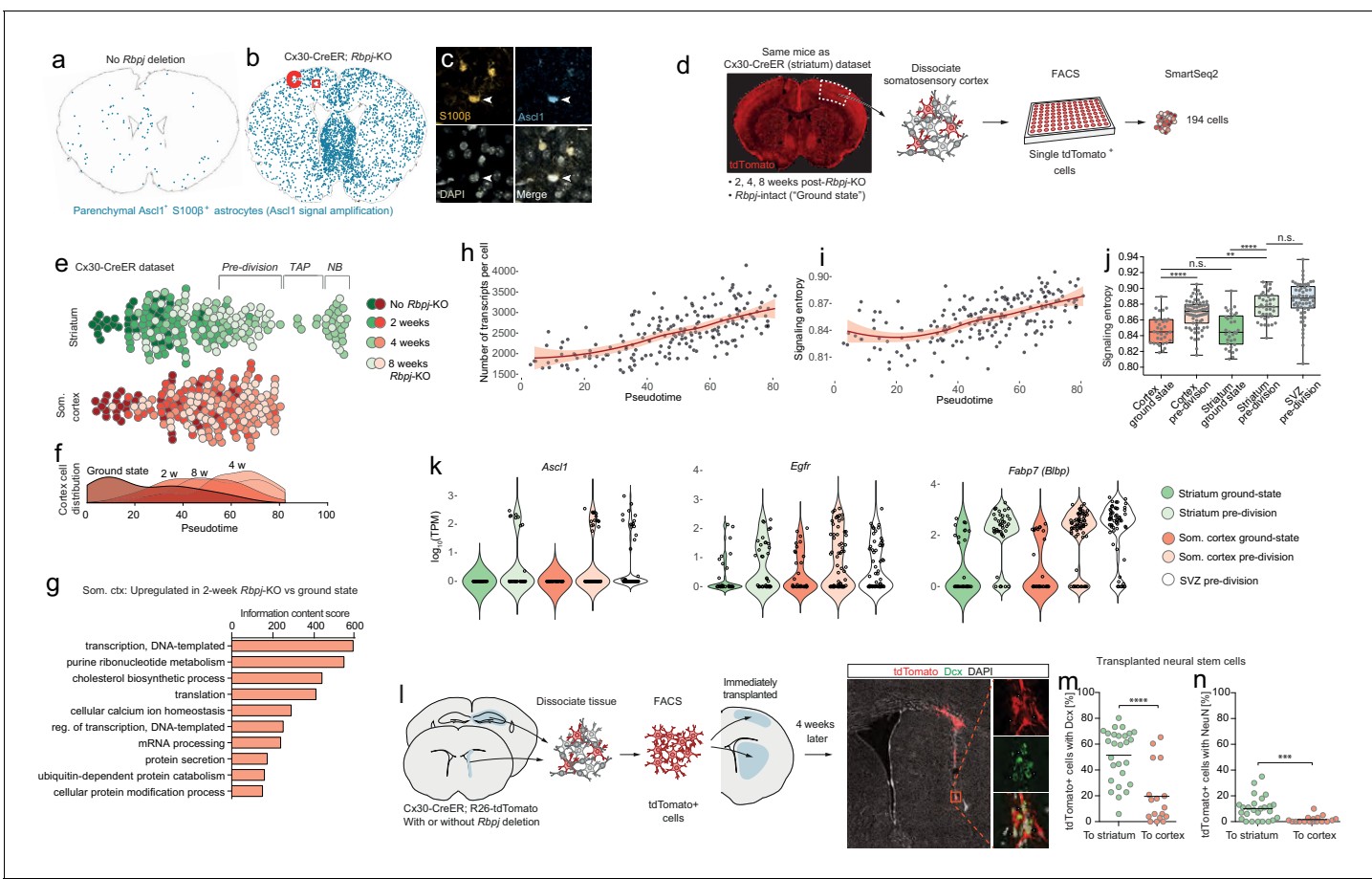

**Figure 5.** Astrocytes outside the striatum initiate an incomplete neurogenic program in response to *Rbpj* deletion. (a–c) Four weeks after *Rbpj* deletion, astrocytes throughout the brain have upregulated low levels of Ascl1 protein, only detectable by using fluorescence-signal amplification (SVZ cells not analyzed or visualized for clarity). (d) We sequenced RNA from somatosensory cortical astrocytes, isolated from the same mice as the striatal cells. (e) We reconstructed the neuronal differentiation trajectory of striatal and somatosensory cortical cells together using Monocle. This analysis reveals that the cortical astrocytes also progress along this trajectory but all stall prior to transit-amplifying divisions. (f) This lineage progression is more clearly visible in a plot showing cell distribution along pseudotime. (g) *Rbpj*-deficient cortical astrocytes upregulate genes primarily related to transcription, translation and lipid metabolism, just like striatal astrocytes. (h) The number of transcripts detected per cell also increases. (i–j) At the same time, their signaling entropy increases to nearly the same levels as in *Rbpj*-deficient striatal astrocytes or SVZ stem cells, indicating an increase in differentiation potency. (k) When cortical astrocytes halt their neurogenic development, they are in their most neural stem cell-like state, having upregulated classical markers of neural stem cells to similar levels as in SVZ stem cells, at least at the RNA level. (l) We isolated neural stem cells from Cx30-CreER; tdTomato mice and immediately transplanted them to litter mates. Four weeks later, a lower proportion of transplanted cells had matured into neuroblasts (m) and neurons (n) in the cortex than in the striatum (each data point represents one recipient mouse). This indicates that the cortical environment is less permissive to neurogenesis than that of the striatum, possibly explaining the failure of astrocytes to complete the neurogenic program there.

The online version of this article includes the following figure supplement(s) for figure 5:

**Figure supplement 1.** Expression levels of neural stem cell markers in striatum, somatosensory cortex and SVZ.

the striatum and cortex. We used Monocle to arrange the cortical cells in pseudotime together with the striatal cells and found that, regardless of brain region, cells from the different time points scored similarly along the initial stages of this pseudotime axis. However, all the cortical astrocytes halted their development prior to entering transit-amplifying divisions (*Figure 5e*). A density plot of cortical astrocytes from the different time points helped better illustrate the gradual progression along this neurogenic axis (*Figure 5f*). Two weeks after *Rbpj* deletion, cortical astrocytes had upregulated genes mainly associated with metabolism and gene expression, just like striatal astrocytes (*Figure 5g*, *Supplementary file 1*). They also steadily increased gene expression activity (*Figure 5h*), again similar to striatal cells. In addition, signaling entropy continuously increased after *Rbpj* deletion (0.85 ± 0.018 vs. 0.87 ± 0.018 [mean ± S.D.] in ground-state vs. pre-division astrocytes, respectively [p<0.0001, unpaired t test]; *Figure 5i–j*) to almost the same level as in the striatal *Rbpj*-deficient astrocytes (*Figure 5j*; pre-division striatal vs. cortical astrocytes: p=0.0091), suggesting that, like the striatal astrocytes, the cortical ones were shifting toward a more stem cell-like state. At their highest pseudotime score, cortical astrocytes were at their most stem cell-like state, characterized by upregulation of neural stem cell genes such as *Ascl1*, *Egfr*, *Fabp7*, *Pou3f4*, *Msi1*, *Slc1a3*, *Cd81* to the same RNA levels as in striatal astrocytes or SVZ stem cells (*Figure 5k* and *Figure 6*). One neural stem cell marker, *Cd9* (*Llorens-Bobadilla et al., 2015*), was conspicuously higher in striatal and SVZ cells than cortical cells (*Figure 5—figure supplement 1*). Another, *Neurog1*, was upregulated by both striatal and cortical astrocytes but was not expressed in the SVZ (*Figure 5—figure supplement 1*). Taken together, it seemed as if cortical astrocytes were preparing to activate the full neurogenic program but that they were unable to enter transit-amplifying divisions.

Cortical astrocytes can be isolated from an injury site and generate neurospheres in vitro (*Shimada et al., 2012*; *Buffo et al., 2008*; *Sirko et al., 2013*). Given this intrinsic capacity to activate neurogenic properties, we therefore wondered whether it might instead be the environment of the cortex that prevents these astrocytes from completing their neurogenic program in vivo. To gauge the neurogenic permissiveness of the striatal and cortical environments, we isolated neural progenitor cells by sorting tdTomato⁺ cells from the SVZ and DG of Cx30-CreER; tdTomato mice (with or without *Rbpj* deletion) 3–6 days after tamoxifen administration (*Supplementary file 3*). These cells were then immediately grafted into the striatum or cortex of littermate mice (*Figure 5l*), to see how many of the grafted cells would form neuroblasts and neurons. When the recipient mice were sacrificed and analyzed 4 weeks later, we found that more of the transplanted cells had developed into Dcx⁺ neuroblasts (*Figure 5m*) and NeuN⁺ neurons (*Figure 5n*) in the striatum than in the cortex (Dcx: striatum 52 ± 21%, cortex 20 ± 22%, mean ± S.D.; NeuN: striatum 10 ± 9%, cortex 2 ± 3%). This suggested that the cortex has an environment less permissive to neurogenesis than the striatum. Such a hostile environment may help explain why *Rbpj*-deficient astrocytes in the cortex fail to complete the neurogenic program that they initiate. Although we focused our analysis on the somatosensory cortex, our immunohistochemical staining of Ascl1 (*Figure 5b*) suggests that an aborted neurogenic program may occur throughout the entire brain in response to *Rbpj* deletion.

## Stalled neurogenic astrocytes in the striatum can resume neurogenesis if exposed to EGF

Although *Rbpj* deletion does trigger neurogenesis by astrocytes in the striatum, this process is most active in the medial striatum (*Figure 1a*). This uneven distribution suggests that in addition to Notch signaling, other mechanisms regulate astrocyte neurogenesis. We wanted to know if our RNA sequencing datasets could help us learn why many striatal astrocytes fail to complete neurogenesis. We noted that, after *Rbpj* deletion, it takes approximately 3–4 weeks until the first striatal astrocytes enter transit-amplifying divisions (*Figure 2c*). Yet, many cells isolated even 8 weeks after *Rbpj* deletion remained as astrocytes that seemed to have halted their development immediately prior to the transit-amplifying cell stage (*Figure 2c*). We asked whether stalled striatal astrocytes could resume their neurogenic program if they were stimulated to enter transit-amplifying divisions. Striatal astrocytes upregulate *Egfr* in response to *Rbpj* deletion (*Figure 3h*). We performed a single injection of the mitogen EGF directly into the lateral striatum of Cx30-CreER; *Rbpj*ᶠˡ/ᶠˡ mice 7 weeks after tamoxifen administration (*Figure 6a*). In *Rbpj*ʷᵗ/ᶠˡ control mice, EGF injection on its own was not enough to induce striatal neurogenesis; neither did it induce any neuroblast migration from the SVZ (*Figure 6b, d*) or any detectable increase in the amount of SVZ neuroblasts (*Figure 6—figure supplement 1a–c*). In wild-type mice, EGF did not trigger cell divisions in striatal astrocytes or non-astrocytes

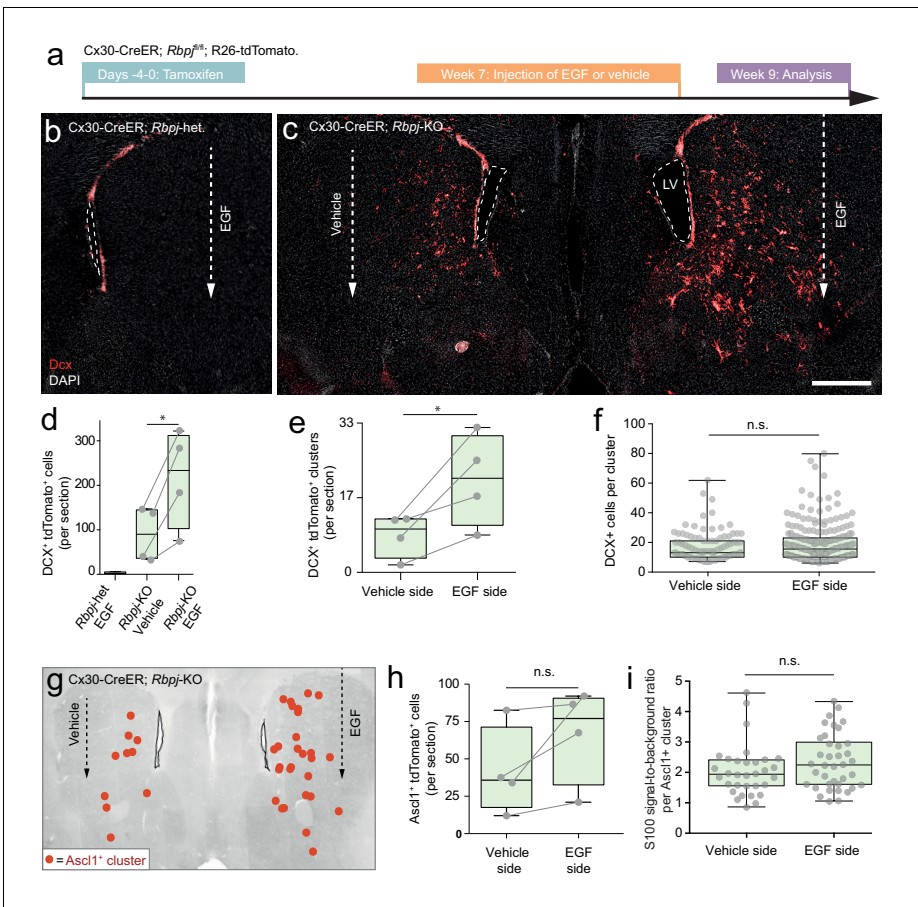

**Figure 6.** Stalled neurogenic astrocytes in the striatum can resume neurogenesis if exposed to EGF. (a) Striatal astrocytes were exposed to EGF delivered through a single injection into the lateral striatum. (b, d) In *Rbpj*-heterozygous control mice, EGF neither causes ectopic migration of SVZ neuroblasts nor neurogenesis by striatal astrocytes. (c, d) However, in *Rbpj* knockout mice, EGF injection leads to a doubling of the number of striatal neuroblasts compared to the vehicle-injected contralateral striatum of the same mice (each data point represents one mouse; lines connect left and right striata from the same mice). (e) EGF injection leads to more striatal neuroblast clusters, but each cluster does not contain more cells (f), indicating that EGF recruits more halted astrocytes into neurogenesis rather than causing increased proliferation by the same number of astrocytes. (g–h) Some mice also show a dramatic increase in the number of transit-amplifying cells; however, this increase was not as pronounced as that of neuroblast clusters. (i) We used residual S100β protein as a short-term lineage tracing marker of striatal astrocyte-derived cells (see *Figure 2—figure supplement 3d–g*). EGF injection does not lead to any influx of SVZ-derived S100β-negative transit-amplifying cells (each dot is one cluster; plot shows sum of three brain sections [n = 3 mice]). Error bars in (d–f, h–i) represent maximum and minimum values. Scale bar (c): 500 μm; LV: lateral ventricle.

The online version of this article includes the following figure supplement(s) for figure 6:

**Figure supplement 1.** EGF injection into the lateral striatum does not change the amount of SVZ neuroblasts.

(*Figure 6—figure supplement 1d*). We concluded that EGF injection into the lateral striatum did not perturb SVZ neurogenesis or trigger confounding striatal proliferation. Interestingly, however, EGF injection into the striatum of *Rbpj*[fl/fl] mice did lead to a doubling in the number of reporter-positive neuroblasts compared to the vehicle-injected contralateral striatum (216 ± 90 [EGF hemisphere, mean ± S.D.], 110 ± 61 [vehicle hemisphere]; p=0.023 [paired t-test]; *Figure 6c–d*). This increase was the result of an increased number of astrocytes entering transit-amplifying divisions rather than increased proliferation by the same number of astrocytes: EGF-injected striata contained on average twice as many neuroblast clusters as vehicle-injected striata (21 ± 10 [EGF hemisphere, mean ±S.D.], 8 ± 5 [vehicle hemisphere]; p=0.049 [paired t-test]; *Figure 6e*), but cluster size did not differ (p=0.59 [Mann-Whitney test]; *Figure 6f*). Some animals also showed a dramatic increase in the number of

Ascl1$^+$ transit-amplifying cell clusters in the EGF-injected striatum (*Figure 6g–h*). This increase, how-ever, was less pronounced (p=0.132 [paired t-test]) than that of Dcx$^+$ cells, possibly because most of the EGF-stimulated astrocytes had already passed the transit-amplifying stage and developed into neuroblasts 2 weeks after the EGF injection. To investigate the local astrocyte origin of these Ascl1$^+$ clusters, we used residual S100β protein as a short-term lineage-tracing marker of striatum-gener-ated transit-amplifying cells (see *Figure 2—figure supplement 3d–g*). We found that S100β protein levels were the same in Ascl1$^+$ clusters in the EGF- and vehicle-injected striata (*Figure 6i*; p=0.159 [unpaired t-test], with a trend toward more S100β$^{high}$ clusters in the EGF-injected striatum). This showed that EGF had caused no influx of S100β-negative cells from the SVZ into the striatum.

Our results demonstrate that EGF administration can enable stalled neurogenic astrocytes in the striatum to initiate transit-amplifying divisions and resume neurogenesis. This effect was easily mea-surable even after a single, localized EGF injection. Yet, striatal neurogenesis was still primarily local-ized to the medial striatum, suggesting that many stalled astrocytes may require a stronger stimulus than a single injection to re-initiate neurogenesis. Further studies would be needed to tell whether longer and more broadly localized EGF exposure could recruit an even larger proportion of stalled striatal astrocytes. EGF exposure in the somatosensory cortex was not sufficient to make stalled *Rbpj*-deficient astrocytes enter transit-amplifying divisions there (*Figure 6—figure supplement 1e*). Interestingly, however, the mere act of inserting a thin syringe needle was enough of a stimulus to do so in some stalled cortical astrocytes (*Figure 6—figure supplement 1e*). We have thoroughly explored the effect of injuries on recruiting halted astrocytes in two separate studies (stab wound in the cortex *Zamboni et al., 2016* and stroke in the striatum *Santopolo et al., 2020*).

Taken together, these results show that it is possible to enhance and modulate the astrocyte neu-rogenic program through targeted interventions, an observation that should be useful when attempting to recruit astrocytes as a reservoir for new neurons throughout the central nervous system.

## Discussion

The brain has almost no capacity to replace lost neurons on its own. Although the SVZ and DG do contain neural stem cells, it is not feasible to use these regions as sources of replacement neurons for the rest of the brain. That is mainly because neurons generated in the SVZ or DG would need to migrate unfeasibly far to reach distant injury sites in the human brain. However, a therapeutic work-around could be to artificially stimulate neurogenesis by cells close to an injury. Astrocytes could be such a reservoir for new neurons because they have an intrinsic neurogenic potential and are very abundant. However, it will be necessary to understand the mechanisms governing astrocyte neuro-genesis before the process can be harnessed and tailored to generate specific neuronal subtypes of significant amounts.

Here, we used single-cell RNA sequencing to study the transcriptional program of parenchymal astrocytes as they initiate neurogenesis in response to deletion of the Notch-mediating transcription factor *Rbpj*. We find that the neurogenic program of parenchymal astrocytes is almost identical to that of SVZ neural stem cells. Based on the analyses in this study, we propose a model where paren-chymal astrocytes are regarded as latent neural stem cells (*Figure 7*).

Specifically, we find that initiation of the neurogenic program in astrocytes is dominated by a steady upregulation of genes associated with transcription and translation, as well as a transient wave of metabolism-associated genes. These findings fit well with what is known about neural stem cell activation in the DG and SVZ, where expression levels of genes associated with transcription and translation steadily increase throughout the neurogenic program (*Llorens-Bobadilla et al., 2015*; *Shin et al., 2015*). Our data suggest that parenchymal astrocytes start off at an even lower baseline level of gene expression than neural stem cells, and that they have increased the number of expressed genes ~ 1.4 fold by the time they reach their most stem cell-like state, prior to transit-amplifying divisions.

Metabolic gene expression peaked right before striatal astrocytes entered transit-amplifying divi-sions, suggesting that the rapid burst of cell divisions is the most metabolically demanding step of the neurogenic program. In the DG, exit from quiescence requires increased lipid metabolism (*Knobloch et al., 2013*), and once activated stem cells enter transit-amplifying divisions, metabolic genes are downregulated, both in the DG and SVZ (*Llorens-Bobadilla et al., 2015*; *Shin et al.,*

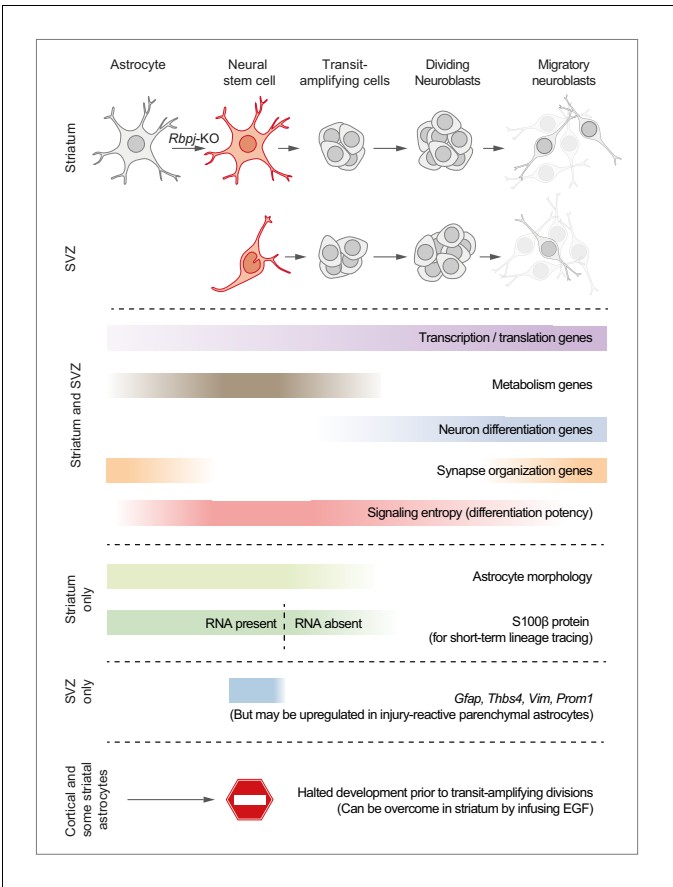

**Figure 7.** Proposed model summarizing the main findings of this study. We propose a model in which striatal astrocytes can be viewed as highly quiescent neural stem cells that, upon *Rbpj* deletion, activate a neurogenic program very similar to that of SVZ stem cells. Some features of the neurogenic program are unique to these parenchymal astrocytes: Their distinct astrocyte morphology is partly retained even after initiation of transit-amplifying divisions (see *Figure 2—figure supplement 3c* and *Video 1*). Furthermore, S100β protein is found in many parenchymal astrocytes, but not in SVZ stem cells, and lingers at low levels in astrocyte-derived transit-amplifying cells even when the *S100b* gene is no longer expressed (dashed line), acting as a short-term lineage tracing marker. Other features are instead unique to the SVZ stem cells, such as expression of *Gfap*, *Thbs4*, *Vim* and *Prom1*, although markers of reactive astrogliosis, such as *Gfap*, are upregulated in neurogenic striatal astrocytes after stroke (*Magnusson et al., 2014*). Astrocytes all over the brain initiate the neurogenic program in response to *Rbpj* deletion, but all astrocytes in the somatosensory cortex, and many in the striatum, halt their development prior to entering transit-amplifying divisions. In the striatum, stalled astrocytes can resume neurogenesis if exposed to EGF.

*2015*). This similarity suggests that, like parenchymal astrocytes, neural stem cells in the DG and SVZ experience a metabolic peak prior to entering transit-amplifying divisions. We find that it takes 2–4 weeks for astrocytes to reach this pre-division state. One possible interpretation of this is that the most time-consuming portion of the astrocyte neurogenic program is the initial exit from deep quiescence, as is the case in quiescent muscle stem cells (*Siegel et al., 2011*). Another (and compatible) interpretation is that *Rbpj* protein may remain bound at its target promoters long after the *Rbpj* gene has been deleted, and that the neurogenic program is initiated only when the protein level has fallen below a certain threshold. Such stable transcription factor-DNA interaction is a mechanism by which some transcription factors buffer against fluctuating transcription levels (*Traets et al., 2020*).

One of our main findings is that all astrocytes in the striatum and somatosensory cortex initiate a neurogenic program in response to *Rbpj* deletion, even though only some astrocytes in the striatum and medial cortex manage to enter transit-amplifying divisions. This suggests that *all* astrocytes in the striatum and somatosensory cortex, and perhaps in other regions as well, are latent neural stem

cells. We hypothesize that the inability of most astrocytes to enter transit-amplifying divisions is due to a lack of proper environmental signals rather than an intrinsic inability. This hypothesis is based on four observations: 1) A large body of literature has demonstrated that parenchymal astrocytes isolated from the injured mouse cortex can generate neurospheres in vitro (*Shimada et al., 2012*; *Buffo et al., 2008*; *Sirko et al., 2013*), proving that astrocytes have intrinsic neural stem cell potential but lack the proper in vivo environment to realize this potential. 2) We here transplanted neural stem cells to the cortex and striatum and found that fewer developed into neuroblasts and neurons in the cortex (*Figure 5l–n*), showing that the cortex has a non-permissive environment. 3) A single injection of EGF into the striatum was enough to make some astrocytes, which had stalled in a pre-division state, enter transit-amplifying divisions (*Figure 6*), showing that extracellular signals can bolster and improve the astrocyte neurogenic program. 4) In Cx30-CreER; *Rbpj*^fl/fl^ mice, most astrocyte-derived transit-amplifying cells and neuroblasts appear in the medial striatum, close to the SVZ. It is possible that the nearby SVZ provides a gradient of neurogenic factors into the medial striatum and establishes a more permissive environment there. Such a gradient has indeed been shown to appear if growth factors are infused into the lateral ventricle (*Yan et al., 1994*; *Anderson et al., 1995*).

In this study, we use *Rbpj* deletion as our trigger to induce neurogenesis by astrocytes. We propose that our results apply also to injury-induced astrocyte neurogenesis even in the absence of *Rbpj* deletion, such as that observed in the stroke-injured striatum. Decreased Notch signaling is the mechanism that triggers neurogenesis by striatal astrocytes in stroke-injured mice (*Magnusson et al., 2014*). Therefore, the *Rbpj* deletion model is a biologically relevant way to activate the astrocyte neurogenic program without triggering confounding tissue processes that occur in response to injury, such as inflammation and reactive astrogliosis. Indeed, the precision afforded by our *Rbpj* deletion model allowed us to observe here that the astrocyte neurogenic program is not dependent on simultaneous reactive gliosis. Instead, neurogenesis and reactive astrogliosis are likely controlled by two independent, but compatible, transcriptional programs.

Is there practical usefulness in designating parenchymal astrocytes as latent neural stem cells? We argue that there is. This nomenclature highlights an aspect of astrocyte biology that exists in parallel to the normal important functions of astrocytes in the healthy brain. Labeling parenchymal astrocytes as latent neural stem cells can help focus research efforts toward activating and fine-tuning the inherent neurogenic capacity of these cells to generate therapeutically useful neuronal subtypes. It is likely that astrocytes in different brain regions use different mechanisms to lock their astrocyte identity in place. Our study demonstrates that it is feasible to identify such mechanisms and seek to target them for enhancing neurogenesis by astrocytes throughout the central nervous system.

## Materials and methods

### Key resources table

| Reagent type (species) or resource | Designation | Source or reference | Identifiers | Additional information |
|---|---|---|---|---|
| Strain, strain background (*Mus musculus,* C57BL/6; M, F) | Cx30-CreER | PMID:17823970 | | |
| Strain, strain background (*Mus musculus,* C57BL/6; M, F) | R26-tdTomato (Ai14) | PMID:25741722 | | |
| Strain, strain background (*Mus musculus,* C57BL/6; M, F) | R26-YFP | PMID:11299042 | | |
| Strain, strain background (*Mus musculus,* C57BL/6; M, F) | *Rbpj*(fl/fl) | PMID:11967543 | | |
| Antibody | anti-Ascl1 (Rabbit polyclonal) | Cosmo Bio | Product No: CAC-SK-T01-003 (RRID:AB_10709354) | Used at 1:2000 |

*Continued on next page*

*Continued*

| Reagent type (species) or resource | Designation | Source or reference | Identifiers | Additional information |
|---|---|---|---|---|
| Antibody | anti-Dcx (Goat polyclonal) | Santa Cruz Biotechnology | Product No: sc-8066 (RRID:AB_2088494) | Used at 1:500 |
| Antibody | anti-GFP (Chicken polyclonal) | Aves Labs | RRID:AB_2307313 | Used at 1:2000 |
| Antibody | anti-Ki67 (Rat monoclonal) | eBioscience | Catalog No: SolA15 (RRID:AB_10853185) | Used at 1:2000 |
| Antibody | anti-S100 (Rabbit polyclonal) | DAKO | Catalog No: GA50461-2 (RRID:AB_2811056) | Used at 1:200 (Directly conjugated to A647) |
| Peptide, recombinant protein | Epidermal growth factor | Sigma-Aldrich | Catalog No: E4127 | |
| Recombinant DNA reagent | AAV8-GFAP(0.7)-iCre-WPRE | Vector Biolabs | Catalog No: VB4887 | |
| Commercial assay or kit | Papain Dissociation System | Worthington | Catalog No: LK003160 | |
| Commercial assay or kit | Adult Brain Dissociation Kit | Miltenyi Biotec | Catalog No: 130-107-677 | |
| Commercial assay or kit | Magnetic MACS Multistand | Miltenyi Biotec | Catalog No: 130-042-303 | |
| Commercial assay or kit | MACS Large Cell Column | Miltenyi Biotec | Catalog No: 130-042-202 | |
| Commercial assay or kit | Myelin Removal Beads II | Miltenyi Biotec | Catalog No: 130-096-731 | |
| Commercial assay or kit | AMPure XT beads | Beckman Coulter | Catalog No: A63881 | |
| Commercial assay or kit | Nextera XT dual indexes | Illumina | | |
| Commercial assay or kit | Chromium Single Cell 3' Library and Gel Bead Kit (v3) | 10X Genomics | Catalog No: CG000183 | |
| Chemical compound, drug | Percoll | GE Healthcare | Catalog No: 17-0891-01 | |
| Chemical compound, drug | EdU | Thermo Fisher | Catalog No: C10424 | |
| Chemical compound, drug | LIVE/DEAD Blue Dead Cell Stain Kit | Thermo Fisher | Catalog No: L23105 | |
| Software, algorithm | R | R Foundation For Statistical Computing | | |
| Other | Agilent Bioanalyzer | Agilent | | |
| Other | BD Influx FACS machine | BD | | |
| Other | 36-gauge beveled needle | World Precision Instruments | Catalog No: NF36BV-2 | |
| Other | NovaSeq 6000 | Illumina | | |
| Other | Illumina HiSeq 2500 | Illumina | | |

## Animals

Cx30-CreER (*Slezak et al., 2007*), R26-tdTomato (Ai14) (*Madisen et al., 2015*), R26-YFP (*Srinivas et al., 2001*), *Rbpj*<sup>fl/fl</sup> *Tanigaki et al., 2002* and wild-type mice were of the C57BL/6 strain

and were >2 months old. Genetic recombination was induced by intraperitoneal tamoxifen injections for five consecutive days (Sigma-Aldrich, St. Louis, MO; 20 mg/ml in 1:9 ethanol:corn oil; 2 mg/injection). Time points were counted from the last day of injection. Mice with no *Rbpj* deletion (for isolation of ground state astrocytes) were sacrificed one day after a three-day tamoxifen regimen. For EdU administration (*Figure 2—figure supplement 3c*), EdU (Thermo Fisher, Waltham, MA) was administered through the drinking water (0.75 g/l EdU in water containing 1% sucrose) for two weeks leading up to the animals' death. Mice were housed in standard conditions with 12/12 hr light/dark cycles and free access to food and water. Experimental procedures were approved by the Stockholms Norra Djurförsöksetiska Nämnd (Permit reference numbers N571-11 and N155-16).

## Immunohistochemistry

Immunohistochemical staining was performed as described previously (*Magnusson et al., 2014*). We used antibodies against Ascl1 (Cosmo Bio, Tokyo, Japan; rabbit, 1:2000), detected with or without ABC kit [Vector Laboratories, Burlingame, CA] and TSA [PerkinElmer, Waltham, MA]), Dcx (Santa Cruz Biotechnology, Dallas, TX; goat, 1:500), GFP/YFP (Aves Labs, Davis, CA, USA; chicken, 1:2000), Ki67 (eBioscience, San Diego, CA, USA; rat, 1:2000), S100 (DAKO, Glostrup, Denmark; rabbit, 1:200, directly conjugated to A647 using Alexa Fluor Antibody Labeling Kit [Thermo Fisher]). EdU was detected using Click-iT EdU Alexa Fluor 647 Imaging Kit (Thermo Fisher).

## Library preparation and sequencing for the Cx30-CreER dataset

One mouse brain was used for each time point, for a total of 4 brains. From each brain, a ~ 1 mm thick coronal slice was isolated, spanning rostrocaudal levels Bregma −0.2 to +1.0 (*Figure 2—figure supplement 1a–c*). From this section, the SVZ was isolated and discarded. The striatum and cortex were then isolated from both hemispheres. Tissue was cut into fine pieces and incubated with papain solution (20 units/ml) containing L-cysteine (1 mM), EDTA (0.5 mM) and DNase (2000 units/ml) (Papain Dissociation System [LK003160], Worthington, Lakewood, NJ), dissolved in EBSS (Thermo Fisher), at 37°C for 30 min, with occasional agitation. Then, the tissue was triturated with smaller and smaller pipet tips and filtered through a 70 µm cell strainer (BD, Franklin Lakes, NJ). After a 10 min centrifugation at 200*g in a swing-bucket rotor at 4°C, supernatant was removed and cells were resuspended in 3 ml sterile-filtered washing buffer (Dulbecco's PBS containing sodium pyruvate, streptomycin sulfate, kanamycin monosulfate, glucose and calcium chloride; Sigma-Aldrich, D4031) containing 0.5% bovine serum albumin (BSA; Sigma-Aldrich, A9418). To remove debris, we then used gradient centrifugation and myelin removal beads in the following way. The cell suspension was carefully layered on top of 4 ml of 90% Percoll (GE Healthcare, Chicago, IL; 17-0891-01, diluted in 1.5 M sterile-filtered NaCl) in a 15 ml Falcon tube and centrifuged at 200*g for 15 min in a swing-bucket rotor at 4°C. After removing the small top phase, the large bottom phase (including the white interphase) was resuspended in 17.5 ml washing buffer in a 50 ml Falcon tube and repeatedly aspirated with a Pipette Boy. This cell suspension was then centrifuged at 200*g for 10 min at 4°C. The pellet was set aside, and to maximize cell yield the supernatant was again repeatedly aspirated and centrifuged. Then, the two pooled cell pellets were resuspended in 1 ml sterile-filtered MACS buffer (5% BSA and 2 mM EDTA in PBS). The cell suspension was incubated with 80 µl myelin removal beads II (Miltenyi Biotec, 130-096-731) for 15 min. Meanwhile, a MACS Large Cell Column (Miltenyi Biotec, Bergisch Gladbach, Germany; 130-042-202) was placed in a Magnetic MACS Multistand (Miltenyi Biotec) and activated by allowing 500 µl of MACS buffer to drop through. The cell suspension was passed through the MACS column and the flow-through was collected in a FACS tube pre-coated with 1% sterile-filtered BSA. MACS buffer (500 µl) was then passed through the column twice to increase cell yield. The cell suspension was then incubated with LIVE/DEAD Blue Dead Cell Stain Kit (Thermo Fisher, L23105) for 30 min. Immediately afterwards, an a BD Influx FACS machine (86 µm nozzle) was used to sort live tdTomato⁺ single cells into 96-well skirted PCR plates (Thermo Fisher) containing lysis buffer (containing dNTP [2.5 mM], oligo-dT [2.5 mM], RNAse Inhibitor [Takara Bio, Kusatsu, Japan; 1:40] and ERCC [Ambion, TX; 1:4,000,000]) with the machine set on 'Index Sort' mode to record cell size information and fluorescence parameters used. Plates were immediately frozen on dry ice and stored at −80°C until processing. For preparation of single-cell cDNA libraries, the plates were thawed at 72°C for 4 min and immediately placed on ice. A reverse transcription of mRNA was performed and the resulting cDNA libraries were immediately subject to

amplification for 19 cycles according to the Smart-Seq2 protocol (*Picelli et al., 2014a*). The final libraries were subjected to bead purification (AMPure, Beckman Coulter, Brea, CA; 0.6:1 bead-to-sample ratio) to remove excess primers and primer dimers. Library quality was checked using an Agilent Bioanalyzer and DNA High Sensitivity microfluidics chips. After determining that libraries were of sufficiently high quality for downstream sequencing, 0.5–1.5 ng of cDNA from each single-cell library was used for tagmentation reactions with a custom Tn5 enzyme/oligo (comparable to the Nextera XT protocol) (*Picelli et al., 2014b*). Nextera XT dual indexes were used (96 index kit) and each library was subjected to an additional 10 cycles of amplification. The final products were again subjected to bead cleaning to remove excess primers and monitored for successful amplification by random sampling of different wells and visualization of library size and quality on the Bioanalyzer. Successful tagmentation reactions were pooled as individual 96-well plates and sequenced using an Illumina HiSeq 2500 machine with 2*125 bp sequencing on High Output mode.

## Read mapping, quality control and computational analyses for the Cx30-CreER dataset

### Read mapping and quality control for striatal cells (Cx30-CreER dataset)

Raw reads were processed with Nesoni clip v. 0.132, removing sequencing adapters and bases with a per-base quality value lower than 20 ('-quality 20') and reads shorter than 64 nucleotides ('-length 64'). Transcript abundance was quantified using Salmon v. 0.8.2 (*Patro et al., 2015*) by pseudo-aligning the filtered reads against the mouse transcriptome (Ensemble release E90; tdTomato sequence was added to the genome) using the cDNA labels combined with the 96 ERCC spike-in controls (Ambion). Median assignment rate was 61% (±10% S.D.). Gene-level summarization was calculated using tximport v. 1.4.0. Cells were sequenced to a median depth of $9*10^5$ (±$4*10^5$ s.D.) reads per cell. Samples were excluded if they had <1500 unique transcripts detected or had a low mapping rate (<20%) or had shown poor library quality when a few samples were analyzed on the BioAnalyzer. Among the remaining cells, we detected a median of 2280 (±883 S.D.) unique transcripts per cell, consistent with previous studies (*Gokce et al., 2016*; *Zeisel et al., 2015*).

### Pseudotime estimation using monocle (Cx30-CreER dataset)

Monocle (v. 1.4.0) was used to generate a pseudotime axis (*Figure 2c*, S5a): Briefly, we first used SCDE (v. 1.99.1) (*Kharchenko et al., 2014*) to generate a list of differentially expressed genes between ground-state and 8 week *Rbpj* deletion astrocytes (genes detected in >10% of cells). Differentially expressed genes present in the GO term Neurogenesis (GO:0022008) were used as input to Monocle. To generate more robust pseudotime values for the cells, we performed the Monocle analysis three times, each time including differentially expressed genes of a different significance level (Z > 1.96, 2.57 or 3.29). Each cell's average pseudotime value was used as the final pseudotime value for making the 'beeswarm plots' (*Figure 2c*). Beeswarm plots were generated using the Beeswarm package (v. 0.2.3).

### Monocle's gene clustering algorithm (Cx30-CreER dataset)

For investigating how groups of genes change as neurogenesis by astrocytes progresses (*Figure 2e–f*, S5b-c), Monocle's gene clustering algorithm was used to group genes with similar expression pattern along pseudotime. First, hierarchical clustering (using hclust [method = ward.d2]) and correlation heatmap plotting of striatal cells (using genes present in the GO term Neurogenesis, and differentially expressed between astrocytes at ground state and 8 weeks after *Rbpj* deletion) revealed five distinct cell groups (*Figure 2—figure supplement 2a*). These groups were sequentially distributed along the neurogenic trajectory in pseudotime (*Figure 2—figure supplement 2b*). We performed differential expression analysis (SCDE) for each of the five groups versus the other four groups to generate five lists of genes particularly highly expressed by each group. The combined gene list was used as input to the gene clustering algorithm.

### Gene ontology analysis (Cx30-CreER dataset)

For gene ontology (GO) analysis, we used DAVID v. 6.8 (*Huang et al., 2009*) and GSAn (*Ayllon-Benitez et al., 2020*). To analyze which subcellular compartments genes belonged to (*Figure 2e*,

S4a), DAVID's GOTERM_CC_FAT option was used. For all other analyses of gene function, the GOTERM_BP_FAT option was used.

## Analysis of SVZ cells from Llorens-Bobadilla et al. (Cx30-CreER dataset)

To compare our Cx30-CreER dataset with that from Llorens-Bobadilla et al., all samples were first mapped together against the mm10 reference transcriptome including the ERCCs using Salmon version 0.8.2. Gene-level TPMs were summarized from the Salmon output files with tximport R package version 1.4.0.

## Assessment of number of genes detected in ground-state astrocytes versus *Rbpj*-deficient astrocytes (Cx30-CreER dataset)

Ground-state astrocytes from the striatum were compared with those from the pre-division stage. Genes with TPM values > 10 were defined as being expressed; the mean expression values from each group's cells were compared using an unpaired t-test.

## Use of cell sorter forward scatter values as a proxy for cell size (Cx30-CreER dataset)

For each sorted cell, a ratio was made between its forward-scatter value and the average forward-scatter value of that sort's tdTomato-negative cell population.

## Sample preparation for the AAV-Cre dataset

Thirteen $Rbpj^{fl/fl}$ mice were subjected to intrastriatal injection of AAV8-GFAP(0.7)-iCre-WPRE virus (VB4887; Vector Biolabs, Malvern, PA, USA) using coordinates in relation to bregma: 1.2 mm anterior, 2.5 mm lateral, 3 mm deep). Five weeks later, mice were sacrificed and the brain collected for isolation of single cells. We used a brain slicer to produce 1.5 mm thick sections spanning the rostro-caudal region transduced by the virus. Prior to isolating striata for tissue dissociation, we inspected fluorescent labeling at the microscope to ensure that only striatal cells (and not SVZ cells) had been transduced. At this stage, two mice were excluded because they had recombined cells in the SVZ. For the remaining 11 mice, striata were microdissected from each slice following the same approach outlined above (sample preparation for the Cx30-CreER experiment), pooled as one sample, and digested using Miltenyi's Adult Brain Dissociation kit. Single cells were finally FAC-sorted based on tdTomato signal as described above (i.e. for the Cx30-CreER dataset). Library preparation was performed using Chromium Single Cell 3' Library and Gel Bead Kit (v3, 10X Genomics, Pleasanton, CA) according to the manufacturer's instructions. Libraries were sequenced on a NovaSeq6000 S1 flow-cell using a custom read setup: 28 nt (read 1), eight nt (i7 index), 166 nt (read 2).

## Pre-processing of the AAV-Cre dataset

Raw reads were obtained from the sequencer and aligned to the mouse genome (mm10) supplemented with tdTomato DNA sequence. We used cellranger v2.0.1 for pre-processing of raw reads followed by Seurat R package (v2.3.4) (*Stuart et al., 2019*) for downstream analysis of mapped data.

Aligned transcripts were normalized for sequencing depth, log-transformed, and scaled using Seurat's default parameters. We excluded genes expressed in less than 3 cells, and cells with mitochondrial content exceeding 25% (n = 418). Next, we ran principal component analysis (PCA) on 2359 highly variable genes and selected the first 10 dimensions for non-linear dimensionality reduction (UMAP) and clustering. The analysis allowed us to identify distinct cell types composing the dataset (*Figure 1—figure supplement 1*). We removed contaminating cells lacking expression of tdTomato (e.g. microglia, n = 87), and cells expressing tdTomato transcripts but not involved in the astroglial neurogenic program (i.e. cells belonging to the oligodendrocyte lineage, n = 1,438). After filtering, we retained 1380 cells for further processing.

## Clustering of astrocytes and their progeny using the AAV-Cre dataset

We included the top 3249 highly variable genes for running PCA and UMAP (using the first 20 PCs) on the filtered dataset. We next performed clustering based on the shared nearest neighbour algorithm (Seurat, v2.3.4) to identify distinct subpopulations within the sample (*Figure 1*). We next, ran pairwise comparisons using Wilcoxon's test to determine cluster-specific transcriptional profiles.

Clustering and differential expression analyses allowed us to record the presence of two astrocyte clusters characterized by expression of *Gjb6* (Astrocyte Cluster 1) and upregulation of *Ascl1* (Astrocyte Cluster 2); a cluster encompassing transit-amplifying cells enriched in *Mki67* transcripts; and two clusters of neuroblasts expressing *Dcx* and *Robo2*, as well as *Foxp2* and *Nav3*. For analyzing differential expression along pseudotime of the AAV-Cre dataset (*Figure 2i*), Monocle (v. 2.12.0) was used.

## Integration with published datasets from SVZ and DG (AAV-Cre dataset)

We aimed at comparing the neurogenic program initiated by striatal astrocytes with that recorded in the neurogenic niches of the SVZ and DG. For this purpose, we compared the transcriptional profiles of cells contained in the current AAV-Cre dataset to that of SVZ (*Zywitza et al., 2018*) (accession code: GSE111527) and DG (*Hochgerner et al., 2018*) (accession code: GSE95315) neurogenesis. From the complete SVZ dataset, we only considered samples collected from wild-type female animals (an002, an003_F, and an003_L). Both SVZ and DG datasets were processed separately, in order to identify distinct cell types and retain only clusters implicated in the neurogenic program. Specifically, we ran PCA on highly variable genes (n = 6497 for SVZ and n = 2224 for DG), followed by tSNE and clustering performed on the top 15 PCs, and selected clusters of cells based on expression of classic neurogenesis-related markers (e.g. *Aqp4*, *Thbs4*, *Sox4*, *Top2a*, *Dcx*) for downstream analysis. Filtered datasets were integrated with striatal data by means of canonical correlation analysis (CCA, Seurat) considering the top 2000 HVGs merged across samples. Datasets were aligned using the first 13 dimensions of CCA and visualized in UMAP space. To investigate the relative distance of cell types across samples we ran hierarchical clustering based on a distance matrix constructed in the aligned space (i.e. using CCA dimensions), followed by differential expression analysis that allowed insight into shared and unique transcriptional programs active in the different neurogenic niches. Finally, we ran correlation analysis on the average expression of each gene to determine similarities in cell-specific transcriptional profiles across brain regions (data were visualized as scatterplots using ggplot 3.3.0).

## Cell transplantation

Donor mice (Cx30-CreER; R26-tdTomato, either with or without inducible *Rbpj* deletion, see *Supplementary file 3*), were sacrificed 3–6 days after intraperitoneal tamoxifen administration. The SVZ and DG were dissected and cells isolated using the same protocol used to isolate cells for the Cx30-CreER dataset. Live tdTomato⁺ cells were sorted using an 86 µm nozzle into a FACS tube pre-coated with 1% BSA, containing 250 µl wash buffer (see cell isolation protocol for the Cx30-CreER dataset). Cell suspensions were kept on ice at all times and chilled during cell sorting. After sorting, cell suspensions were transferred to pre-coated microcentrifuge tubes and centrifuged at 200*g for 10 min in a swing-bucket rotor at 4°C. The swing-bucket rotor was important as it enabled the small cell pellet to collect at the bottom of the tube. Supernatant was removed carefully. Tubes were then briefly centrifuged in a tabletop centrifuge to bring down a few extra microliters of fluid from the walls of the tube. This amount of liquid was enough to resuspend the cell solutions, which was done by carefully flicking the tubes. If more volume needed to be added, wash buffer was used. Cell suspension was transferred to 0.2 ml tubes, placed on ice and immediately transplanted to the brains of littermate mice. One microliter of this solution was injected into each of the following coordinates (relation to bregma): Striatum: 0.5 mm anterior, 2.0 mm lateral, 3.0 mm deep. Cortex: 0.5 mm anterior, 2.5 mm lateral, 1.0 mm deep. The entire protocol took ~16–18 hr to perform from beginning to end. During the analysis of transplantation-recipient mice, mice were excluded if <20 transplanted cells were found.

## EGF injection

In Cx30-CreER; *Rbpj*^fl/fl and *Rbpj*^+/fl mice, intrastriatal EGF injection was performed seven weeks after tamoxifen administration. EGF (Sigma-Aldrich E4127, 200 ng/µl) was dissolved in PBS containing 0.1% bovine serum albumin (BSA). Four Cx30-CreER; *Rbpj*^fl/fl mice and four Cx30-CreER; *Rbpj*^fl/wt control mice were anesthetized with isoflurane (4% at 400 ml/min flow rate to induce anesthesia, followed by ~2% at 200 ml/min). Then, 2 µl of EGF solution was delivered into the lateral striatum

through a single injection (0.7 mm anterior, 2.4 mm lateral of bregma; 3 mm deep of dura mater) using a 36-gauge beveled needle (World Precision Instruments, Sarasota, FL, USA). A vehicle solution consisting of 0.1% BSA in PBS was injected into the contralateral striatum of the same mice. Mice were sacrificed and analyzed 2 weeks after EGF injection. For estimating whether EGF induces proliferation in wild-type mice (*Figure 6—figure supplement 1d*), mice were sacrificed 48 hr after EGF injection.

Evaluation of EGF's effect on cortical astrocytes was evaluated as follows. Animals were first injected with 0.3 µl of AAV-GFAP-Cre into the somatosensory cortex to delete *Rbpj* locally (1.0 mm anterior, 1.5 mm lateral of bregma. Liquid was injected at both 1.0 and 0.5 mm deep of dura mater). Both left and right hemispheres were injected with the virus. Four weeks later, 2*0.3 µl of EGF solution was delivered into one hemisphere using the same injection coordinates. A vehicle solution consisting of 0.1% BSA in PBS was injected into the other hemisphere of the same mouse. Mice were sacrificed and analyzed 2, 3 and 4 weeks after EGF injection.

### Evaluation of the effect of EGF on the number of SVZ neuroblasts

To estimate the number of neuroblasts in the SVZ (*Figure 6—figure supplement 1*), we performed an immunohistochemical staining against Dcx and used Dcx signal intensity as a proxy for cell numbers. We took tiled images of the SVZ in 3–5 brain sections per mouse using a Zeiss LSM 700 confocal microscope, using the same microscope settings for the left and right SVZ. CellProfiler v. 3.0.0 (*Carpenter et al., 2006*) was used to identify Dcx$^+$ cells or cell clusters and calculate total signal intensity (cell area*mean intensity) on the EGF- and vehicle-injected hemispheres in all images.

### Assessment of S100β protein levels in striatal Ascl1$^+$ cells for use as a short-term lineage tracing marker

To detect residual S100β protein levels in striatal Ascl1$^+$ cells, we took images of all Ascl1$^+$ cells found in the striatum and SVZ of five brain sections of one mouse, adjusting imaging depth for each cell and using the same microscope settings for every cell. Then, we measured Ascl1 and S100β signal intensity in each cell using ImageJ's Measure function (*Schneider et al., 2012*).

### Calculation of signaling entropy to estimate differentiation potency

We used LandSCENT v. 0.99.3 (*Chen and Teschendorff, 2019*) to calculate signaling entropy on the single-cell level. Because the LandSCENT package only contains a protein-protein interaction network for human genes, we first converted our dataset's mouse gene names into their human orthologous counterparts using biomaRt. Only genes that were expressed (TPM $\geq$10) in $\geq$10% of cells were included in the analysis.

## Acknowledgements

We want to thank Nigel Kee and Joanna Hård for valuable discussions, and Michael Ratz for help with preparing libraries for the AAV-Cre dataset. We thank Sarantis Giatrellis and Marcelo Toro for flow cytometry. This study was supported by the Swedish Research Council, the Swedish Cancer Society, SSF, the ERC, the Knut and Alice Wallenberg Foundation and Torsten Söderberg Foundation.

## Additional information

### Funding

| Funder | Grant reference number | Author |
| --- | --- | --- |
| Svenska Forskningsrådet Formas | | Jonas Frisén |
| Cancerfonden | | Jonas Frisén |
| Stiftelsen förStrategisk Forskning | | Jonas Frisén |

| H2020 European Research Council | ERC-2015-AdG - 695096 | Jonas Frisén |
| --- | --- | --- |
| Knut och Alice Wallenbergs Stiftelse | | Jonas Frisén |
| Torsten Söderbergs Stiftelse | | Jonas Frisén |

The funders had no role in study design, data collection and interpretation, or the decision to submit the work for publication.

## Author contributions

Jens P Magnusson, Conceptualization, Data curation, Software, Formal analysis, Supervision, Validation, Investigation, Visualization, Methodology, Writing - original draft, Project administration, Writing - review and editing, Designed the study, performed experiments, analyzed results, performed computational analyses of the Cx30-CreER dataset and prepared the manuscript; Margherita Zamboni, Data curation, Software, Formal analysis, Validation, Investigation, Visualization, Methodology, Writing - review and editing, Performed experiments, analyzed results, interpreted data, took part in experimental design and contributed to editing the manuscript. Performed the computational analyses of the AAV-Cre dataset; Giuseppe Santopolo, Data curation, Formal analysis, Validation, Investigation, Visualization, Methodology, Writing - review and editing, Performed experiments, analyzed results, interpreted data, took part in experimental design and contributed to editing the manuscript; Jeff E Mold, Data curation, Formal analysis, Investigation, Methodology, Prepared sequencing libraries for the Cx30-CreER dataset; Mauricio Barrientos-Somarribas, Carlos Talavera-Lopez, Software, Mapped sequencing reads; Björn Andersson, Supervision, Supervised CT-L and MB-S; Jonas Frisén, Conceptualization, Resources, Supervision, Funding acquisition, Methodology, Writing - original draft, Project administration, Writing - review and editing, J.F. performed project design and coordination, interpreted data and prepared the manuscript

## Author ORCIDs

Jens P Magnusson ⓘ https://orcid.org/0000-0002-3928-8959
Margherita Zamboni ⓘ https://orcid.org/0000-0003-0664-4707
Jonas Frisén ⓘ https://orcid.org/0000-0001-5819-458X

## Ethics

Animal experimentation: All animal experimental procedures were approved by the Stockholms Norra Djurförsöksetiska Nämnd (Permit reference numbers N571-11 and N155-16).

## Decision letter and Author response

Decision letter https://doi.org/10.7554/eLife.59733.sa1
Author response https://doi.org/10.7554/eLife.59733.sa2

## Additional files

### Supplementary files

• Source data 1. Raw data for plots.

• Supplementary file 1. Genes differentially expressed between ground-state and *Rbpj*-deficient astrocytes. Data is shown for both striatum and somatosensory cortex. Astrocytes at the pre-division stage are in their most stem cell-like state. Therefore, genes differentially expressed at this time point represent potential marker genes for the astrocyte stem cell state.

• Supplementary file 2. Lists of genes characteristic for the four gene clusters in *Figure 2e*. Results from a GO analysis (DAVID) is provided for each gene cluster.

• Supplementary file 3. Details about mice transplanted with neural stem cells (*Figure 6l–n*). This list provides information about the mice that were used as transplantation donors (see also Methods).

• Transparent reporting form

## Data availability

The Cx30-CreER dataset (fastq files and processed expression matrix) has been deposited in ArrayExpress (accession E-MTAB-9268). The AAV-Cre dataset has been deposited in the Gene Expression Omnibus (GEO; accession GSE153916).

The following datasets were generated:

| Author(s) | Year | Dataset title | Dataset URL | Database and Identifier |
|---|---|---|---|---|
| Magnusson JP, Frisén J, Zamboni M, Santopolo G, Mold JE, Barrientos-Somarribas M, Talavera-Lopez C, Andersson Br | 2020 | Activation of a neural stem cell transcriptional program in parenchymal astrocytes | https://www.ebi.ac.uk/arrayexpress/experiments/E-MTAB-9268/ | ArrayExpress, E-MTAB-9268 |
| Zamboni M, Magnusson JP, Frisén J | 2020 | Transcriptomics analysis of neurogenesis by striatal astrocytes upon Rbpj-K deletion | http://www.ncbi.nlm.nih.gov/geo/query/acc.cgi?acc=GSE153916 | NCBI Gene Expression Omnibus, GSE153916 |

The following previously published datasets were used:

| Author(s) | Year | Dataset title | Dataset URL | Database and Identifier |
|---|---|---|---|---|
| Llorens-Bobadilla E, Zhao S, Baser A, Saiz-Castro G, Zwadlo K, Martin-Villalba A | 2015 | Single-Cell Transcriptomics Reveals a Population of Dormant Neural Stem Cells that Become Activated upon Brain Injury | https://www.ncbi.nlm.nih.gov/geo/query/acc.cgi?acc=GSE67833 | NCBI Gene Expression Omnibus, GEO:GSE67833 |
| Zywitza V, Misios A, Bunatyan L, Willnow TE, Rajewsky N | 2018 | Single-cell transcriptomics characterizes cell types in the subventricular zone and uncovers molecular defects impairing adult neurogenesis. | https://www.ncbi.nlm.nih.gov/geo/query/acc.cgi?acc=GSE111527 | NCBI Gene Expression Omnibus, GEO:GSE111527 |
| Hochgerner H, Zeisel A, Lönnerberg P, Linnarsson S | 2018 | Conserved properties of dentate gyrus neurogenesis across postnatal development revealed by single-cell RNA sequencing | https://www.ncbi.nlm.nih.gov/geo/query/acc.cgi?acc=GSE95753 | NCBI Gene Expression Omnibus, GEO:GSE95753 |

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
