## [Decision Letter]

**Acceptance summary:**

The manuscript studies the mechanisms by which specialized astrocytes in distinct anatomic regions, triggered by stoke and decreased notch signaling, initiate a neurogenic program. The two main findings are that: 1.as astrocytes initiate neurogenesis they first become transcriptionally similar to SVZ progenitors; 2. astrocytes that do not progress to a progenitor state can be induced to do so by administration of EGF in vivo.

**Decision letter after peer review:**

Thank you for submitting your article "Activation of a neural stem cell transcriptional program in parenchymal astrocytes" for consideration by *eLife*. Your article has been reviewed by three peer reviewers, and the evaluation has been overseen by Joseph Gleeson as the Reviewing Editor and Marianne Bronner as the Senior Editor. The following individuals involved in review of your submission have agreed to reveal their identity: Jianwei Jiao (Reviewer #2); Sebastian Jessberger (Reviewer #3).

The reviewers have been given the opportunity to discuss the reviews with one another and the Reviewing Editor has drafted this decision to help you prepare a revised submission.

Summary:

The manuscript studies the mechanisms by which specialized astrocytes in distinct anatomic regions, triggered by stoke and decreased notch signaling, initiate a neurogenic program. *Rbpj*, a Notch-mediating transcription factor, is used to modulate Notch signaling with temporal and spatial control. This is a very well executed study that looks at understanding the neurogenic potential of astrocytes in vivo. The authors use scRNAseq to profile genetically traceable astrocytes that have been "induced" to reprogram into neurons. The authors show that deletion of *Rbpj* induces molecular programs that resemble expression profiles of endogenous, neurogenic NSCs (from the SVZ) when *Rbpj* is deleted in striatal areas (more so in the medial parts of the striatum) that are permissive for neurogenesis. In contrast, they find that cortical astrocytes upon *Rbpj* deletion initially enter a NSC-like program but become stalled before differentiating into a transit-amplifying state. Two distinct labeling methods are used and the work is very extensive in its characterization. There are two main findings: 1. That as astrocytes initiate neurogenesis they first become transcriptionally similar to SVZ progenitors; 2. That astrocytes that do not progress to a progenitor state can be induced to do so by administration of EGF (in vivo).

Essential revisions:

– Can the authors provide a better molecular definition of the astrocyte-derived progenitors. It is clear that they cluster closer to SVZ progenitors, but what are their core genetic signatures (I imagine they will be 30-40 genes)?

– In the induction (by EGF) experiment, how many astrocytes can now enter the trans-amplifying progenitor stage? Is this a tissue-level effect? What happens to other cells around the astrocytes in response to EGF.

– The serial numbers in some figure legends are confused and misleading. The authors should consider to put the numbers in front of each sentence rather than at the end of sentence and keep them in series.

– The description of these figures should be short and clear, which will be easier to read. I suggest that some legend details are better to write in the main text.

– Figure 5C seems not the regions labeled in B. The signal pattern is different.

– For Figure 1A, authors need to provide some details about how they analyze DCX+ signal and the standard they used to represent DCX+ with red dots in the Materials and methods.

– Of course, it would be interesting to see if also cortical astrocytes can be induced to overcome a pre-TAP roadblock by infusion of EGF. This somewhat obvious experiment is missing – at the same time it is not absolutely critical to support the main conclusions of the data.

---

## [Author Response]

Essential revisions:– Can the authors provide a better molecular definition of the astrocyte-derived progenitors. It is clear that they cluster closer to SVZ progenitors, but what are their core genetic signatures (I imagine they will be 30-40 genes)?

We thank the reviewers for the great suggestion of providing a more detailed molecular definition of striatal astrocytes in their stem cell-like state. We now include a list of all genes differentially expressed between striatal astrocytes at ground state versus their stem cell-like pre-division state. This list is included as a tab in Supplementary file 1 and is referred to in the main text (subsection “Parenchymal astrocytes are located upstream of a neural stem cell state and acquire a transcriptional profile of neural stem cells upon their activation”). This gene list provides a detailed view of which genes have been both up- and downregulated compared to ground state astrocytes, which may provide both a better understanding for the astrocyte neurogenic program as well as putative marker genes for the astrocyte stem cell state. For example, one conspicuous group of genes that emerge as highly specific to the astrocytes’ stem cell state are multiple paralogs of the serine-protease inhibitor Serpina3 (Serpina3c, Serpina3f, Serpina3k, Serpina3m, Serpina3n).

– In the induction (by EGF) experiment, how many astrocytes can now enter the trans-amplifying progenitor stage?

The reviewers ask a very interesting question: How many astrocytes can now enter the transit-amplifying stage, i.e. *are now capable* of doing so? Our scRNA-seq data suggest that all astrocytes are *capable* of entering transit-amplifying divisions. Our EGF experiment shows that even a single injection of EGF is enough to double the number of astrocytes that do so. That number is still relatively small: In each brain section analyzed, we find evidence that on the order of tens of striatal astrocytes per brain section have entered transit-amplifying divisions at the time point of our analysis (Ascl1+ clusters plus Dcx+ clusters). There are on the order of 500-1000 striatal tdTomato+ astrocytes per brain section and hemisphere (data not shown). Thus, most *Rbpj*-KO astrocytes had not entered transit-amplifying divisions at the time point of our analysis. However, our EGF administration was highly localized and of very short duration.

The reviewers’ question can also be interpreted to mean: “If EGF could be administered with optimal concentration, duration and distribution within the striatum, would EGF be capable of recruiting every single stalled *Rbpj*-KO astrocyte into transit-amplifying divisions, or would some astrocytes still be stuck in their stalled state?” Unfortunately, the results from our experiment cannot address this highly interesting question. We have now added a sentence in the manuscript (subsection “Stalled neurogenic astrocytes in the striatum can resume neurogenesis if exposed to EGF”) to mention this point.

Is this a tissue-level effect?

Our experiment does not provide evidence that EGF induces a tissue-wide re-initiation of astrocyte neurogenesis. We administered EGF very locally using stereotaxic surgery. Even so, the distribution of neurogenic astrocytes remained quite similar as before, with more neurogenic cells in the medial striatum. This suggests that, although EGF manages to push many striatal astrocytes into transit-amplifying divisions, additional mechanisms are likely to regulate this step, and may be identified in the future. We have included a sentence in the manuscript (subsection “Stalled neurogenic astrocytes in the striatum can resume neurogenesis if exposed to EGF”) to clarify this point.

What happens to other cells around the astrocytes in response to EGF.

We now include a panel in Figure 6—figure supplement 1, and a sentence in the main text (subsection “Stalled neurogenic astrocytes in the striatum can resume neurogenesis if exposed to EGF”) to show what happens to both astrocytes and non-astrocytes 48 hours after a striatal EGF injection in wild-type mice. Neither astrocytes nor non-astrocytes show any increased cell division after EGF injection. Figure legends and the Materials and methods have been updated to reflect these additions.

– The serial numbers in some figure legends are confused and misleading. The authors should consider to put the numbers in front of each sentence rather than at the end of sentence and keep them in series.

This has now been done.

– The description of these figures should be short and clear, which will be easier to read. I suggest that some legend details are better to write in the main text.

We have now shortened the figure legends.

– Figure 5C seems not the regions labeled in B. The signal pattern is different.

Thank you for pointing this out. The square in Figure 5B has now been adjusted.

– For Figure 1A, authors need to provide some details about how they analyze DCX+ signal and the standard they used to represent DCX+ with red dots in the Materials and methods.

This is a relevant point. Because Dcx is a microtubule-binding protein, it shows a filamentous staining pattern that can stretch quite far from the cell nucleus. However, the red dots in Figure 1A represent only the nuclei of Dcx+ cells. (By the way, the same is true for all other Dcx quantifications in the study) Dcx+ cells always show a very bright Dcx signal, so there was no need to define signal-to-noise cutoff levels. We have now specified in the figure legend of Figure 1A that red dots represent nuclei of Dcx+ cells.

– Of course, it would be interesting to see if also cortical astrocytes can be induced to overcome a pre-TAP roadblock by infusion of EGF. This somewhat obvious experiment is missing – at the same time it is not absolutely critical to support the main conclusions of the data.

EGF injection did not show the same effect in the somatosensory cortex as in the striatum. We now include these data as a panel in Figure 6—figure supplement 1 and an explanatory sentence in the main text (subsection “Stalled neurogenic astrocytes in the striatum can resume neurogenesis if exposed to EGF”). Interestingly, however, just inserting a thin syringe needle into the cortex was sufficient to push some cortical *Rbpj*-KO astrocytes into transit-amplifying divisions. We have thoroughly explored the effect of injury in recruiting astrocytes to neurogenesis in two separate studies and found that a stab wound injury in the cortex as well as a striatal stroke promotes this process when Notch signaling is blocked. We now include references to these studies in the aforementioned subsection (Zamboni et al., in press; Santopolo et al., 2020).